



# Fjord circulation induced by melting icebergs

Kenneth G. Hughes[1]

[1]College of Earth, Ocean, and Atmospheric Sciences, Oregon State University, Oregon, USA

**Correspondence:** Kenneth Hughes (kenneth.hughes@oregonstate.edu)

**Abstract.**

In an iceberg-choked fjord, meltwater can drive circulation. Down-fjord of the ice, buoyancy and rotation lead to an outflowing surface coastal current hugging one side of the fjord with an inflowing counter current below. To predict the structure and evolution of these currents, we develop an analytical model—complemented by numerical simulations—that involve a rectangular fjord initially at rest. Specifically, we (i) start with the so-called Rossby adjustment problem, (ii) reconfigure it for a closed channel with stratification, and (iii) generalize the conventional 'dam-break' scenario to a gradual-release one that mimics the continual, slow injection of meltwater. Implicit in this description is the result that circulation is mediated by internal Kelvin waves. The analytical model shows that if the total meltwater flux increases (e.g., a larger mélange, warmer water, or enhanced ice–ocean turbulence), then circulation strength increases as would be expected. For realistic parameters, a given meltwater flux induces an exchange flow that is ∼50 times larger. This factor decreases with increasing water column stratification and vice versa.

## 1 Introduction

'Nowhere in the sea could a melting iceberg be expected to have a more pronounced effect on its environment than in the enclosure of a fjord' (Gade, 1979). In Greenland, many fjords house hundreds or thousands of icebergs. For example, at any given time Sermilik Fjord is home to $\mathscr{O}(10\,000)$ icebergs, and their cumulative freshwater flux of $\sim500\,\mathrm{m^3\,s^{-1}}$ is equivalent to a moderately large river (Moon et al., 2018; Moyer et al., 2019; Rezvanbehbahani et al., 2020).

The obvious consequences of iceberg melt are cooling and freshening of near-surface waters. Less obvious are the consequences for circulation. As Enderlin et al. (2016) noted, fjord circulation studies 'have



largely ignored the contribution of iceberg melt'. Similarly, Beaird et al. (2017) noted a lack of work on the impact of 'distributed buoyancy forcing on fjord circulation' in Ilulissat Icefjord. Davison et al. (2020) addressed the issue by developing a model of Sermilik Fjord that parameterized iceberg thermodynamics at the sub-grid scale. They predicted that the outflow over the top $\sim$200 m is a few cm s$^{-1}$ over the top
$\sim$200 m, and that the compensating inflow over the 200–500 m depth range increases advection of warm Atlantic Water toward the glaciers at the head of the fjord.

It is easy to intuit the idea that meltwater is buoyant and therefore rises up and out of the fjord at the surface and that a compensating inflow is needed at depth. But this does not help with more quantitative questions. Why are the currents the speeds they are? What controls the depths of the inflow and outflow?
What would happen in wider or narrower channels? How would results change with different stratification or a different melt parameterization? These questions were not the focus of the study by Davison et al. (2020) nor subsequent ones that use the same iceberg parameterization (Davison et al., 2022; Kajanto et al., 2023; Hager et al., 2023).

The core contribution of this paper is an analytical model explaining the first-order dynamics of a
fjord's response to hundreds of melting icebergs. To gain process-level insight and make the problem tractable, we use a semi-realistic approach. For example, the fjord has a realistic width and depth but is idealized as a rectangle. And the icebergs have realistic sizes, but are distributed such that there is a clear line separating mélange and open water. Further, we ignore other forcings like subglacial discharge and shelf-driven baroclinic flow, and we investigate the initial value problem of a fjord starting from rest. These
simplifications let us best illustrate the role of waves in setting the circulation.

Before presenting the analytical model, we develop and run a high-resolution numerical model of the same semi-realistic scenario (Section 2). This provides a specific realization, or *answer*, that we then reverse-engineer. After developing analytical model in ten steps (Section 3), we test its skill by against the numerical model (Section 4), and then use it in a parameter space study (Section 5).





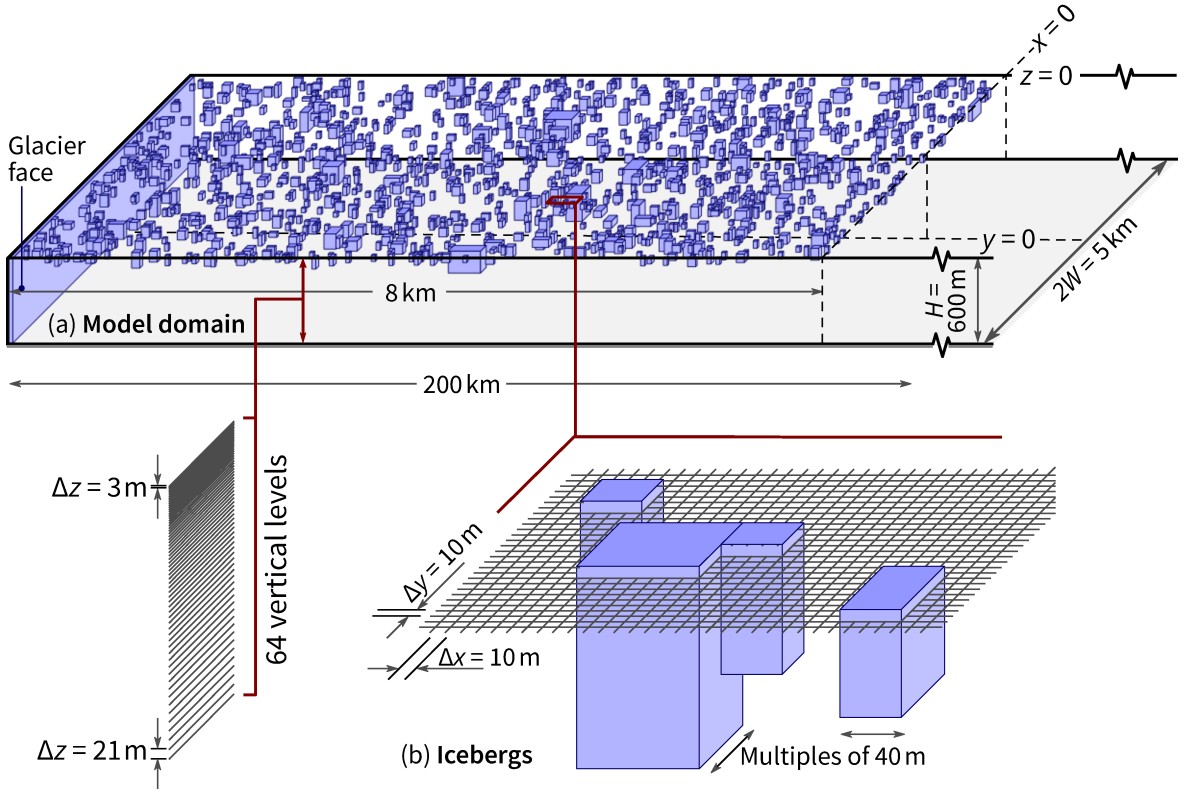

**Figure 1.** Numerical model setup. (a) A long rectangular fjord contains a $8\,\text{km} \times 5\,\text{km}$ mélange in which 10% of the surface area is covered by icebergs. Note the coordinate system with $x = 0$ at the end of the mélange, $y = 0$ in the fjord center, and $z = 0$ at the surface. (b) Volume-occupying icebergs are explicitly resolved in the model.

## 2 Numerical modeling

### 2.1 Numerical model setup

We simulate the dynamics of a rectangular fjord that is $600\,\text{m}$ deep ($H$) and $5\,\text{km}$ wide (half-width $W = 2.5\,\text{km}$) using the MITgcm (Marshall et al., 1997; Adcroft et al., 2004). The domain extends $200\,\text{km}$ in the along-fjord (east–west) direction, and in the first $8\,\text{km}$ there is a mélange that comprises $\sim$1200 icebergs covering 10% of the ocean surface (Figure 1a).

Melting ice is the only forcing. The icebergs, which are cuboids, extend over many grid cells horizontally and act as upside-down topography (Losch, 2008). Melting on the base and sides cools and and freshens the adjacent ocean cells, with ice–ocean thermodynamics calculated using the three-equation formulation





(Appendix A). The cuboids follow a power-law size distribution. The number of icebergs of a given

horizontal area goes as $A^{-1.9}$ (Sulak et al., 2017), which means that smaller icebergs are more numerous than larger ones. Horizontally, iceberg dimensions are rounded to multiples of 40 m and a maximum of 320 m×240 m is imposed. Vertically, most icebergs have a keel depth of 30–80 m. Further details and justification of the fixed cuboid approach are given by Hughes (2022).

The ambient water initially has a constant potential temperature and linear salinity profile, and hence

constant buoyancy frequency:

$$S_a(z) = 35\,\text{psu} - \Delta S\left(1 + \frac{z}{H}\right) \tag{1}$$

$$\theta_a(z) = 2\text{°C} \tag{2}$$

$$N = \sqrt{g\beta\frac{\Delta S}{H}} \tag{3}$$

where $\Delta S = 3\,\text{psu}$ is the salinity increase between the surface and the seafloor, which is representative

of values of 2–4 observed in Greenland fjords (e.g., Straneo et al., 2011; Sciascia et al., 2013), and $\beta \approx 7.8 \times 10^{-4}\,\text{psu}^{-1}$ is the saline contraction coefficient. The 2°C value equates to approximately 4°C above the freezing, an average thermal forcing for Greenland fjords (Wood et al., 2021). Constant temperature precludes local warming at mid depths caused by upward advection of warm water (cf. Davison et al., 2022).

The coordinate system has $x = 0$ at the end of the mélange, $y = 0$ in the center, and $z = 0$ at the surface. Within the mélange ($x < 0$), $\Delta x = \Delta y = 10\,\text{m}$. Outside the mélange ($x > 0$), $\Delta x$ increases 3% per cell. There are 64 vertical levels, with the highest resolution of $\Delta z = 3\,\text{m}$ at the surface. The time step is 2 s and simulations are run in hydrostatic mode for one week. Vertical mixing is parameterized with the Klymak and Legg (2010) overturning scheme with a background viscosity and diffusivity of $A_v = K_v = 10^{-4}\,\text{m}^2\,\text{s}^{-1}$.

Horizontal viscosity is parameterized with a Smagorinksy viscosity with the `viscC2Smag` coefficient set to 2.5 together with a background value of $A_h = 10^{-3}\,\text{m}^2\,\text{s}^{-1}$. Temperature and salinity are advected with a third-order flux limiter scheme (designated as scheme 33 in the MITgcm).

The Coriolis parameter is set for 70°N ($f = 1.37 \times 10^{-4}\,\text{s}^{-1}$). The channel width $2W = 0.6L_R$, where $L_R = c_1/f = 8.6\,\text{km}$ is the mode-1 internal Rossby radius. The mode-1 internal wave speed $c_1$ has a

realistic value of $1.2\,\text{m\,s}^{-1}$ (e.g., Sutherland et al., 2014)





## 2.2 Numerical model results

The simulated icebergs melt at 0.3–0.4 m d$^{-1}$, which is at the upper end of typical observed values (Enderlin and Hamilton, 2014; Enderlin et al., 2018; Schild et al., 2021). This melt creates flows of order 5 cm s$^{-1}$ (Figure 2). The fastest flows occur in (i) near-surface hotspots where currents are squeezed through gaps between icebergs and (ii) the coastal current downstream at $x > 20$ km. This current hugging the 'right-hand side' is the most obvious consequence of Coriolis.

Most of the surface flow within the mélange is down-fjord as expected. The westward flow in the southwest corner (Figure 2a) is perhaps unintuitive and will be explained in Section 3. When averaged over the whole mélange region, the outflow is down-fjord in the top 50 m and up-fjord below that (Figure 2e). Notably, this up-fjord flow occurs despite an appreciable release of meltwater down to 100 m.

Averaged over the week-long simulation, freshwater and heat fluxes are 110 m$^3$ s$^{-1}$ and 36 TW. By the end of the week, the near-surface salinity and temperature have fallen by ∼0.3 psu and ∼1°C (Figure 2g–h).

Starting from rest, the coastal current down-fjord of the mélange is the first major feature to form (Figure 3). As with the averaged flows described earlier, the coastal current flows down-fjord in the top 50 m and up-fjord over the 50–150 m range. Within 12 h, the current reaches $x = 10$ km (Figure 3b). This is too fast to be explained by advection because the currents themselves are only 1–3 cm s$^{-1}$. Instead, a faster wave mechanism is at play. This mechanism also explains why the coastal current extends only small fraction across the channel despite the channel width being only $0.6L_R$. Specifically, the decay away from the wall is faster than $\exp(-(y+W)/L_R)$ because the current has a structure comprising many modes, not just mode 1, and the higher modes have shorter Rossby radii.

Advection *is* responsible for the increasingly long channel-wide jet. Between $t = 2$ and 4 days, the easternmost end of the jet moves from $x = 2$ to 7 km (Figure 3d–e) at 3 cm s$^{-1}$. However, except for this jet, the evolving system that we have simulated numerically can be explained and quantified in terms of wave mechanics with an analytical model.

## 3 Analytical modeling

We will build the core of our analytical model in nine steps (Sections 3.1–3.9). Each step is based on wave mechanics in a certain scenario for which its section is named. The role of melt is added as a tenth step.



**Figure 2.** Dynamics of the top half of the fjord after one week of simulation. A simple description is of surface outflow in the top 50 m and inflow at 50–150 m, but Coriolis and advection complicate the picture. (a, b) A plan view. (c) An along-fjord slice. (d) A cross-fjord slice. (e–h) Properties averaged or integrated over the mélange region.

The first three sections consider the Rossby adjustment problem in a channel (also called geostrophic adjustment). We start with a two-dimensional problem of flow generated by an abrupt release of a region of high sea surface height in an open channel (Section 3.1), which is the simplest case mathematically



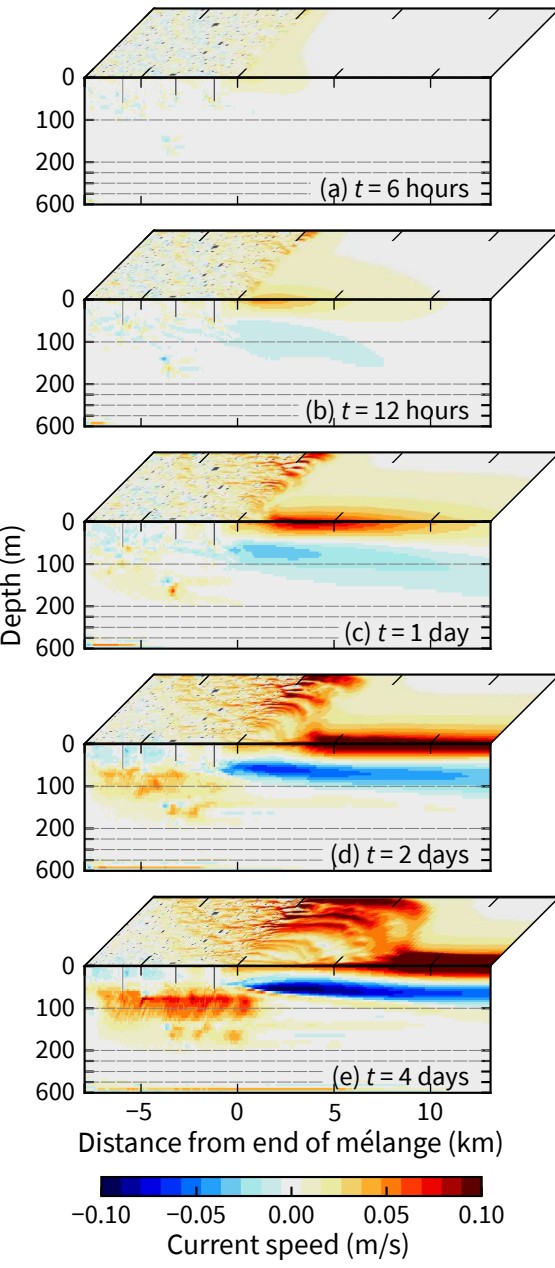

**Figure 3.** Development of fjord circulation from rest as depicted by vertical and horizontal slices at the southernmost side and surface, respectively ($y = -W$ and $z = 0$). The vertical axes are enlarged in the top 200 m where currents are fastest.




and conceptually. We then make the geometry more realistic by closing one end (Section 3.2). Finally, we change the forcing for this closed case from abrupt to gradual (Section 3.3), which is a better analogue for a melting ice system.

The middle three sections consider a different two-dimensional problem: baroclinic adjustment without
rotation in the $x - -z$ domain. Again, we first consider abrupt releases in open and closed settings (Sections 3.4 and 3.5, respectively) and then look at the closed case with gradual forcing (Section 3.6)

The last three sections consider the three-dimensional problem with rotation. For the third time, we start with open and closed settings with abrupt releases (Sections 3.7 and 3.8, respectively) and then the closed setting with gradual forcing (Section 3.9)

Note that in Sections 3.1–3.3, 'downstream' will be to the left because it will simplify Figure 4. In all other sections, downstream or 'down-fjord' will be to the right.

### 3.1 Barotropic, open channel, abrupt release, rotating

A sea surface height discontinuity in a fluid at rest is unbalanced. When released, the discontinuity generates waves and currents that restore equilibrium. Here we consider this evolution toward equilibrium for a
rotating, shallow water system in a channel. Kelvin and Poincaré waves are generated, and the former leave behind steady boundary currents with *e*-folding scales equal to the Rossby radius.

Consider a single discontinuity perpendicular to the channel axis (Figure 4a). Fluid initially accelerates from higher to lower sea surface height. On a time scale of $\mathcal{O}(f^{-1})$, this down-channel flow turns to the right (northern hemisphere) to form a cross-channel jet centered about the original discontinuity. On the
wall where the jet converges, a Kelvin wave of elevation is generated; on the opposite wall, a Kelvin wave of depression. Over time, these two waves propagate away and leave behind two coastal currents moving in the same direction, but on opposite sides of the channel. The two currents are connected by the original cross-channel jet.

Hermann et al. (1989) give analytical expressions for this *wave-adjusted state* (i.e., the linear solution
that sets up before slower advective dynamics develop). Well downstream of the initial discontinuity



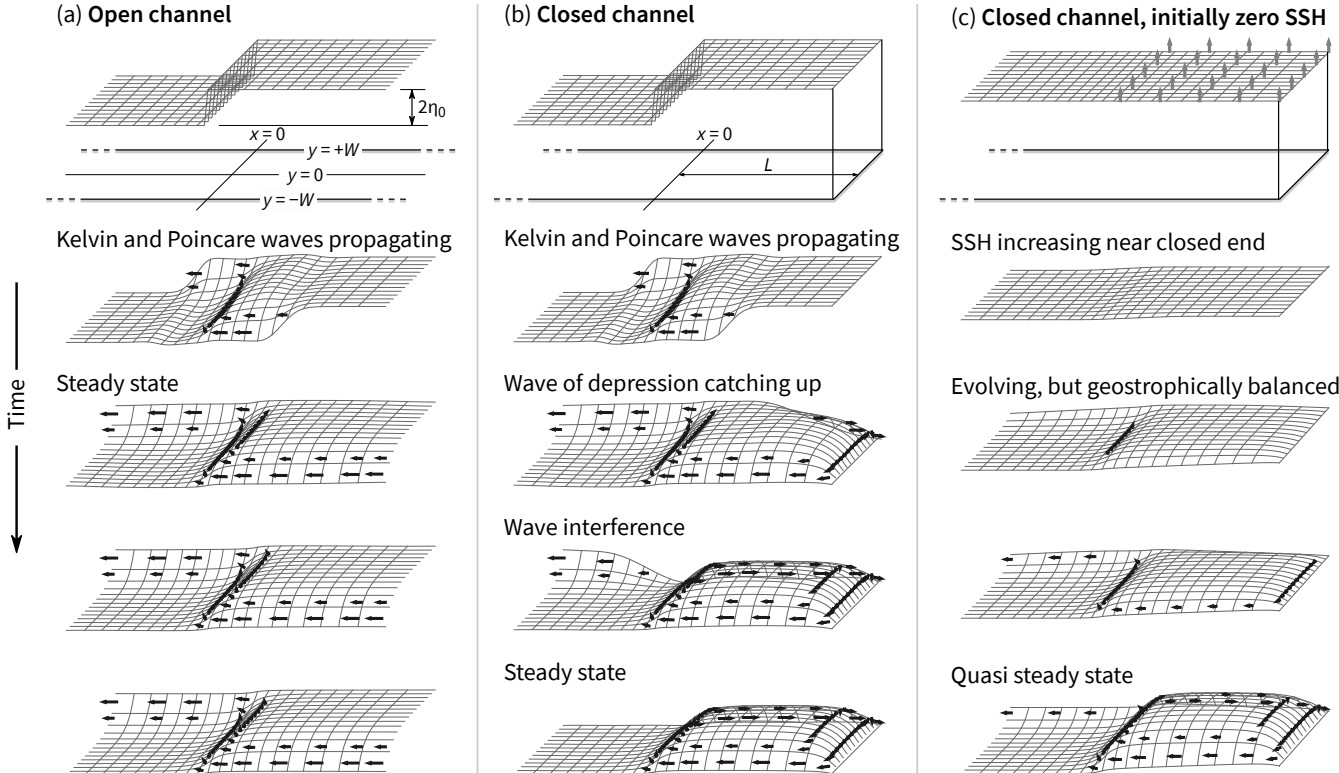

**Figure 4.** Rossby adjustment in a channel for three different scenarios. (a) The conventional problem with a sea surface height discontinuity in the middle of an open channel. (b) The same as panel a, but with one end of the channel closed. (c) The same as panel b, but with the sea surface being continually pushed upward from zero rather than having an initial discontinuity. Height anomalies are exaggerated in all panels.

$(x \ll 0)$, the sea surface height $\eta$, along-channel velocity $u$, and total down-channel flux $Q$ are

$$\eta = \eta_0 \left[ 1 - \mathrm{sech}(W/L_R) \exp(y/L_R) \right] \tag{4}$$

$$u = \eta_0 \sqrt{g/H} \, \mathrm{sech}(W/L_R) \exp(y/L_R) \tag{5}$$

$$Q = -2\eta_0 \sqrt{gH} \tanh(W/L_R) L_R \tag{6}$$

where $\eta_0$ is half height of the initial surface discontinuity, $W$ is the half width of the channel, and $H$ is the depth. Hermann et al. (1989) also provide expressions for $u$ and $v$ close to the discontinuity, but these are more elaborate and overkill for our purposes.





Evaluating Equation 5 at the boundary gives

$$u(x \ll 0, y = +W) = \eta_0 \sqrt{g/H}(1 + \tanh(W/L_{\mathrm{R}}))  \tag{7}$$

For an infinitely narrow channel, the tanh term goes to zero and gives the non-rotating limit. For a wide
channel ($W \gg L_{\mathrm{R}}$), the tanh term goes to one and the maximum velocity is double the non-rotating limit.
Regardless of channel width, the cross-channel integral of the change in potential energy is equal to the
cross-channel integral of kinetic energy:

$$\int 0.5g(\eta - \eta_0)^2 \mathrm{d}y = \int 0.5u^2 H \mathrm{d}y = g\eta_0^2 L_{\mathrm{R}} \tanh(W/L_{\mathrm{R}})  \tag{8}$$

## 3.2   Barotropic, closed channel, abrupt release, rotating

If the channel is closed on the high surface end as in Figure 4b, then the wave of depression cannot
propagate toward $x = \infty$. Instead, the wave propagates around the closed boundary and—after turning
two corners—starts propagating in the same direction and on the same side of the channel as the wave of
elevation. When this second wave passes back beyond $x = 0$, it starts to cancel the effects of the first. Hence,
for $x < 0$, the down-channel flux is nonzero only between the arrival times of the two waves. The extra
distance that the wave of depression travels is $2L + 2W$, so a time series of the flux is a top-hat function
with nonzero values over a period $\Delta t$ where

$$\Delta t = (2L + 2W)/c  \tag{9}$$

## 3.3   Barotropic, closed channel, gradual forcing, rotating

In Figure 4c there is no initial discontinuity. Instead the sea surface is continually pushed upward in the
$x > 0$ region. This upward movement can be treated as a sum of sequential infinitesimal perturbations.
Hence, the system's response can be treated as a sum of sequential infinitesimal solutions (assuming
linearity). In contrast to the abrupt closed case, $\eta, u$, and $Q$ are nonzero in the $x < 0$ region for all values of
$t$. Expressions for these three quantities are the same as Equations 4–6, but with

$$\eta_0 \rightarrow \frac{\mathrm{d}\eta_0}{\mathrm{d}t}\Delta t  \tag{10}$$

where $\mathrm{d}\eta_0/\mathrm{d}t$ is the rate at which the sea surface is pushed upward and $\Delta t$ is the wave delay from Equation 9.



### 3.4 Baroclinic, open channel, abrupt release, non-rotating

We turn now to a depth dependent scenario: a non-rotating, open channel in two dimensions ($x$–$z$) with the left half having a low-density anomaly near the surface (Figure 5a). In practice, the density anomaly would vary with depth as a function of meltwater input; for now, we impose a constant anomaly in the top third of the water column.

The density field of the final, steady state is easy to predict. Assuming no mixing occurs, the final state must contain the same fluid parcels as the initial state, but sorted vertically such that no available potential energy remains. Hence, the final density field is horizontally homogeneous (Figure 5e).

Provided the initial density anomaly is small relative to the background stratification, it will *not* slump as if it were a gravity current (e.g., Simpson, 1982). Rather, each unstable fluid parcel only travels a short vertical distance before it reaches its neutral buoyancy level. Indeed, throughout this paper we are assuming that the vertical movement of meltwater is limited by stratification and hence vertical scales are a few tens of meters at most (e.g., Huppert and Turner, 1980; Yang et al., 2023).

The rearranging of fluid parcels produces pressure perturbations that lead to internal waves spreading out in both directions from the location of the initial density discontinuity (Figure 5b–d). For constant stratification $N$ and a fjord depth $H$, the speed of a mode-$n$ internal wave is

$$c_n = \frac{NH}{n\pi} \tag{11}$$

Throughout the main text, we are using constant stratification for its simplicity. The generalization of the analytical model to any stratification profile is described in Appendix B.

The two mode-1 waves spread out rapidly from the initial discontinuity. On the right-hand side, there is a wave with positive vertical anomalies (green shades in Figure 5b–d). On the left-hand side, there is a corresponding wave with negative vertical anomalies. The horizontal velocity induced by these waves is the same on either side: eastward flow near the surface and westward flow deeper down. For points between the mode-1 and mode-2 wavefronts, the velocity profile is

$$u(z) = A_{u_1} \cos\left(\frac{\pi z}{H}\right) \quad \text{for } c_2 t < |x| < c_1 t \tag{12}$$





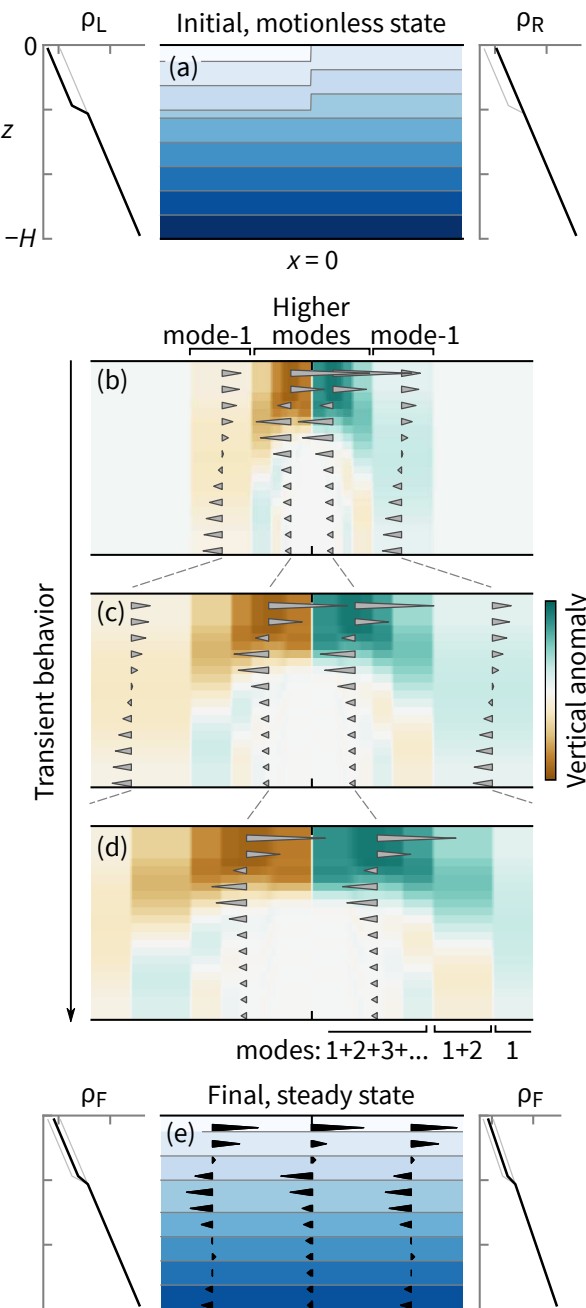

**Figure 5.** Baroclinic adjustment in a stratified fluid of a density anomaly near the surface of a non-rotating, open channel. The available potential energy in the initial, motionless state is all converted to the kinetic energy of the final, steady state. When the initial pressure anomaly is considered as a sum of internal modes, the final state—and the evolution toward it—can be predicted from the known behavior of those modes.





where $A_{u_1}$ is a Fourier coefficient to be determined later. Once the mode-2 wavefront passes the same location, its velocity superimposes on that already there. Hence,

$$u(z) = A_{u_1} \cos\left(\frac{\pi z}{H}\right) + A_{u_2} \cos\left(\frac{2\pi z}{H}\right) \quad \text{for } c_3 t < |x| < c_1 t \tag{13}$$

Similarly, the vertical anomaly has both mode 1 and 2 components (see, e.g., the location labeled 'mode 1+2' in Figure 5d)

In the long-time limit, the velocity at any horizontal location is given by the generalization of Equation 13:

$$u(z) = \sum_{n=1}^{\infty} u_n(z) = \sum_{n=1}^{\infty} A_{u_n} \cos\left(\frac{n\pi z}{H}\right) \tag{14}$$

In the example in Figure 4, the final velocity is 17% mode-1, 25% mode-2, 22% mode-3, 12% mode-4, and 24% higher modes.

The coefficients $A_{u_n}$ can be derived from the initial density profiles. To start, define the density anomaly $\rho'$ as

$$\rho'(z) = (\rho_R - \rho_L)/2 \tag{15}$$

where $\rho_L$ and $\rho_R$ are the density profiles for the initial state on the left- and right-hand sides. Then, integrate this with depth to get a hydrostatic pressure anomaly $p'$ that is the same on both the left- and right-hand sides:

$$p'(z) = \frac{g}{\rho_0} \int_z^0 \rho'(z^*) z^* \mathrm{d}z^* \tag{16}$$

where $z^*$ is a dummy variable used to avoid ambiguity with the integral's lower limit. The value of $p'(z)$ is
zero at the surface and negative below.

Like $u(z)$, the profile $p'(z)$ can be described as a cosine series:

$$p'(z) = \frac{A_{p_0}}{2} + \sum_{n=1}^{\infty} A_{p_n} \cos\left(\frac{n\pi z}{H}\right) \tag{17}$$

where

$$A_{p_n} = \frac{2}{H} \int_{-H}^{0} p'(z) \cos\left(\frac{n\pi z}{H}\right) \tag{18}$$





There is a simple link between the coefficients $A_{u_n}$ and $A_{p_n}$:

$$A_{u_n} = C n A_{p_n}. \tag{19}$$

Therefore,

$$u(z) = C \sum_{n=1}^{\infty} n A_{p_n} \cos\left(\frac{n\pi z}{H}\right) \tag{20}$$

If $\rho'(z)$ is spread out with depth, then $p'(z)$ and $u(z)$ are shaped more by low mode components. If $\rho'(z)$ is surface intensified, then $p'$ and $u$ have higher mode components (Figure 6).

The proportionality constant $C$ in Equation 20 is found implicitly by equating the available potential energy $E_\mathrm{p}$ of the initial state to the kinetic energy $E_\mathrm{k}$ based on the velocity in the final state $u_F$, where

$$E_\mathrm{p} = \frac{g^2}{2\rho_0} \int_{-H}^{0} \frac{\rho'^2}{N^2} \, dz \tag{21}$$

$$E_\mathrm{k} = \frac{\rho_0}{2} \int_{-H}^{0} u_F^2 \, dz \tag{22}$$

Equation 21 follows from what Kang and Fringer (2010) call APE$_3$. It is a good approximation here because vertical perturbations are small.

### 3.5 Baroclinic, closed channel, abrupt release, non-rotating

Baroclinic adjustment in a closed channel follows the same steps as for an open channel, but the boundary breaks the symmetry and means that no steady state arises. Consider the scenario in Figure 7a of a near-surface, low-density anomaly of width $L$ beside the closed end, where $L$ is much smaller than the channel length. Initially, the transient behavior of the system matches that in the open channel case in Figure 5b: for each mode, two waves spread out from $x = 0$. After reaching the boundary and reflecting back to $x = 0$, the originally westward waves destructively interfere with their counterparts. Because the eastward waves had an effective headstart of distance $2L$, the velocity field tends toward a series of stripes, each of width $2L$ (Figure 7b). The lower, faster modes are further to the right; the higher, slower modes trail behind. (Note the difference in presentation between Figure 5, which shows velocities as arrows, and Figure 7, which shows velocities with a red–blue colormap.)



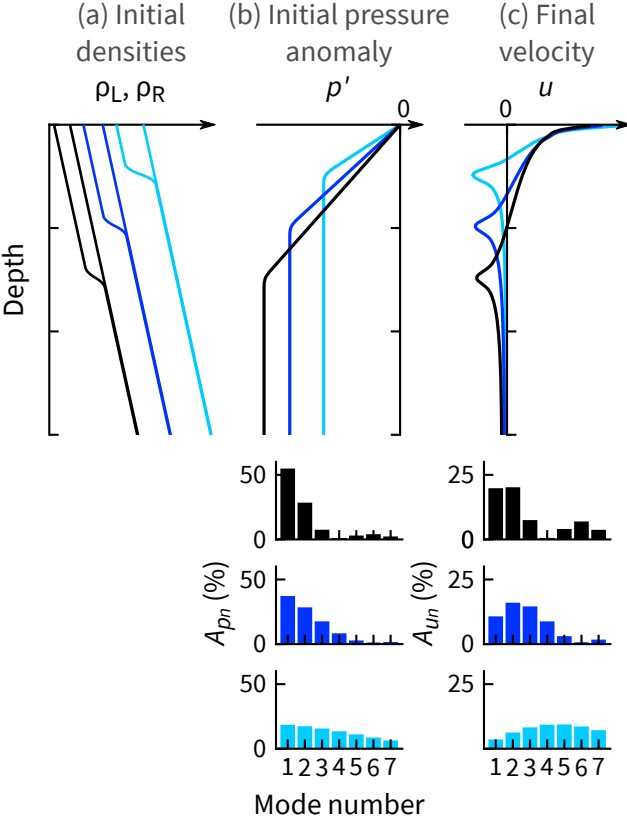

**Figure 6.** Examples of the link between $p'(z)$ and $u(z)$ (Equations (14) and (18)) for baroclinic adjustment of a non-rotating, stratified system. The black example has an initial density discontinuity that is most spread out with depth, so $p'(z)$ has relatively low-mode components. Hence, the associated $u(z)$ also has relatively low-mode components (Equation 19). Conversely, the light blue example is more surface-intensified and has relatively high-mode components. In all cases, the initial available potential energy is the same.

The potential and kinetic energy in the closed case are similar to Equations (21) and (22), but because of the loss of symmetry, both expressions need to be integrated in the $x$ direction to ensure energy is globally 240 conserved:

$$E_{\mathrm{p}}(t=0) = \frac{g^2 L}{\rho_0} \int\limits_{-H}^{0} \frac{\rho'^2}{N^2}\, \mathrm{d}z \tag{23}$$

$$E_{\mathrm{k}}(t=\infty) = \rho_0 L \sum_{n=1}^{\infty} \int\limits_{-H}^{0} u_n^2\, \mathrm{d}z \tag{24}$$



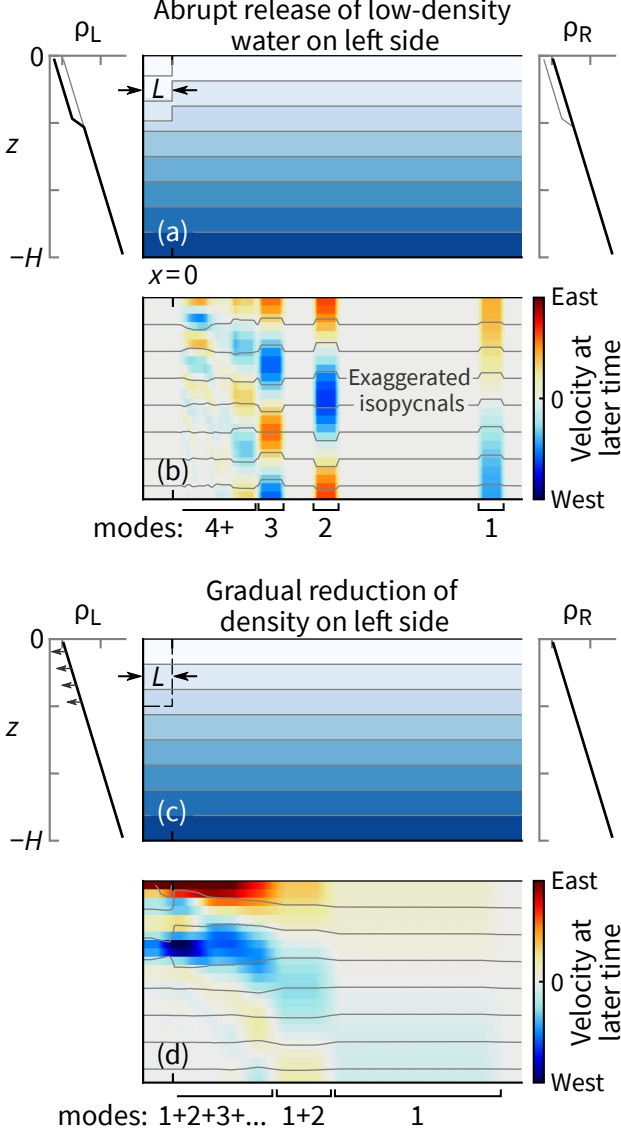

**Figure 7.** Baroclinic adjustment of density anomalies near the closed end of a stratified channel. (a–b) When released abruptly, the density anomaly generates two waves for each mode that spread out in each direction from $x = 0$, just as in Figure 5. However, the originally westward waves reflect at the closed end and destructively interfere with the eastward waves, except for the $2L$-wide portion of the waves that had a headstart. (c–d) If the density anomaly is gradually built up from nothing, then the outcome is the sum of sequential and infinitesimal versions of the solution in panel b.




The potential energy expression comes from Equation 21 multiplied by $2L$ where $L$ comes from integrating in $x$ and the factor of 2 occurs because the equilibrium state for $\rho(z,t)$ is now $\rho_R$, not $(\rho_L + \rho_R)/2$ as it was for the open case. The kinetic energy expression, which involves $u_n$ as defined in Equation 14, is best understood from Figure 7b: the total kinetic energy is the sum of all the $2L$-wide bands of nonzero velocity, each of which are a single mode provided they have had time to sufficiently spread out from each other. Indeed, the kinetic energy of the system increases monotonically in time and only approaches Equation 24 as $t \to \infty$.

The velocity field at any given time, depth, and position (assuming $x > 0$) can be summarized as the total velocity induced by the eastward-propagating waves minus that of the waves that initially propagated westward and then reflected:

$$u(x,z,t) = \sum_{n=1}^{n_E} u_n(z) - \sum_{n=1}^{n_W} u_n(z) \tag{25}$$

The integer $n_E$ is the maximum $n$ for which $c_n t > x$, and similarly for $n_W$ for $c_n t > x + 2L$.

## 3.6  Baroclinic, closed channel, gradual forcing, non-rotating

Consider now the same closed system, but with the low-density anomaly gradually added (Figure 7c). As in Section 3.3, we can treat this as a series of sequential and infinitesimal versions of the closed system that we just solved. For example, let the density everywhere be $\rho_R$ at $t = 0$ and let the left side reduce at a constant rate $(\rho_L - \rho_R)/\tau$. In other words, $\tau$ is the time it would take for the left-hand side to reach $\rho_L$ assuming the system was held in place so that it could not respond.

If we want to know the velocity field at any time $t$, then we sum a large number $m$ (e.g., 100) solutions to the abrupt-release, closed case (Equation 25) that are evenly spaced in time up until $t$. Specifically,

$$u(x,y,z,t) = \sum_{i=1}^{m} u(x,y,z,it/m) \tag{26}$$

with the effective $\rho_L$ reduced accordingly

$$\rho_L \to \rho_R - \frac{t}{m\tau}(\rho_R - \rho_L) \tag{27}$$

For example, if $m = 100$, then we effectively sum 100 abrupt-release cases that start at $0.00t, 0.01t, 0.02t, \ldots, 0.99t$, with each having values of $\rho_R - \rho_L$ that are scaled down by a factor of 100. Although we could also express this more formally in the limit $m \to \infty$, doing so does not provide extra insight.





With this gradual release in a stratified channel, we can start to see a resemblance between the analytical
and numerical models (e.g., compare Figure 7d to Figure 3c).

### 3.7   Baroclinic, open channel, abrupt release, rotating

Three-dimensional baroclinic Rossby adjustment in a channel combines concepts from the $x$–$y$ and $x$–$z$
cases from Sections 3.1–3.6. We will use the same initial density field as in Sections 3.4–3.6 and make it
constant in the across-channel direction (Figure 8a).

When the density discontinuity is released, it generates two counter-propagating families of Kelvin
waves (Figure 8b): westward ones on the far side of the channel ($y = +W$) and eastward ones on the closer
side ($y = -W$), just as occurred in the simpler form with barotropic waves in Figure 4a. Individually, each
mode behaves like the barotropic case, except with a depth-dependent velocity and a much reduced Rossby
radius. The latter for a given mode $n$ is

$$L_{Rn} = \frac{NH}{n\pi f} \tag{28}$$

which follows from the wave speed expression in Equation 11. Mode-1 Rossby radii are typically 5–10 km
in Greenland fjords.

For regions away from the initial discontinuity, the velocity is parallel to the walls and the expression for
$u(x,y,z,t)$ is separable in three dimensions. In $x$ and $z$, we use the two-dimensional baroclinic solution from
Section 3.4. In $y$, velocities decay exponentially and follow the scaling in Equation 5. More specifically,
the velocity field is

$$u(x,y,z,t) = \sum_{n=1}^{n_E} A_{u_n} \cos\left(\frac{n\pi z}{H}\right) \text{sech}(W/L_{Rn}) \exp(-\text{sgn}(x)\,y/L_{Rn}) \tag{29}$$

where $A_{u_n}$ are the velocity coefficients derived for the non-rotating $x$–$z$ solution, and $n_E$ is defined in
Section 3.5.

A curious consequence of the superposition in Equation 29 is that, at certain points, $u$ does not decay
monotonically away from the boundary. Instead, the differing scales and phases of the individual modes
can lead to local maxima or minima in $u(y)$ within the channel. (However, a clear example of this does not
arise in Figure 8b).



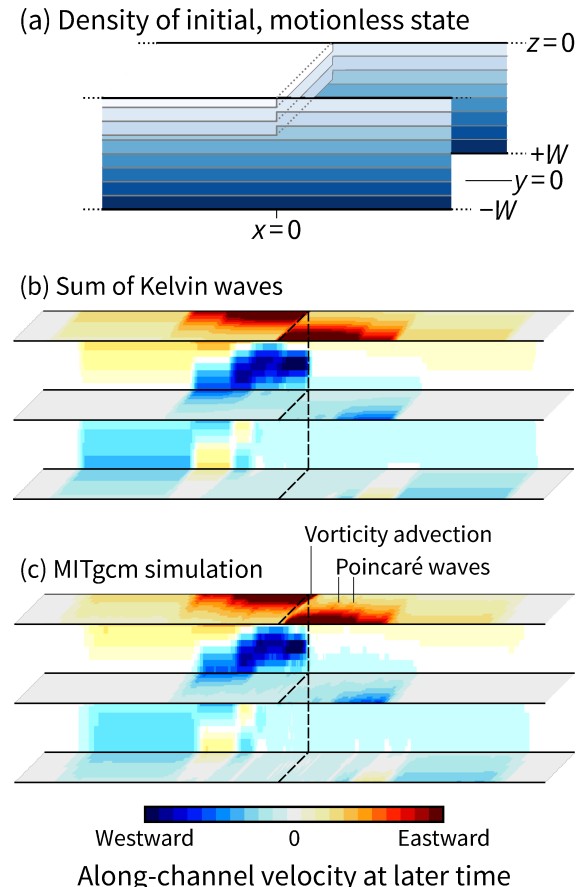

**Figure 8.** Baroclinic Rossby adjustment in an open-ended, stratified channel predicted by analytical and numerical models. The values of velocity are unimportant here, but the initial densities and the colormaps for panels b and c are identical. The MITgcm simulation shows two slight additions that do not arise with a semi-geostrophic approximation: advection of the discontinuity and Poincaré waves. For this particular example, the channel width is equal to the mode-1 internal Rossby radius.





To confirm that the sum of Kelvin waves is a good approximation of the true system, we test it against

a MITgcm simulation with the same initial conditions. (This simulation is separate to those described in

Sections 2 and 4.) In Figure 8, panels b and c clearly agree, and the same is true for wider and narrower

channels (not shown). The MITgcm simulation does contain additional physics—vorticity advection and

Poincaré waves—but the effect on along-channel velocity is small. (See Hutter et al. (2011) for further

details on Kelvin and Poincaré waves in channels of constant depth.)

Not shown in Figure 8 are the cross-channel velocities. It would be possible, albeit cumbersome, to

derive $v(x,y,z,t)$ from linear wave dynamics as we have done for $u(x,y,t,z)$; the starting point would be the

solution for the barotropic case (Equation 2.22 of Hermann et al. (1989)). From the MITgcm simulations,

we find (as expected) that $v$ peaks near $x = 0$ and goes to zero at $|x| \gg 0$ (not shown). It is negative at the

surface: water flows from $y = +W$ to $-W$. A counterflow from $-W$ to $+W$ occurs at the depth $z = -H/3$

where the discontinuity goes to zero.

### 3.8   Baroclinic, closed channel, abrupt release, rotating

In a closed channel, the baroclinic Kelvin waves propagating toward the closed end start at $(x,y) = (0,+W)$,

travel a distance $L$ westward, turn a corner, travel a distance $2W$ southward, turn another corner, and travel

a distance $L$ eastward to arrive at $(0, -W)$. Thereafter, they destructively interfere with their counterparts.

This is the same argument as in the barotropic case in Section 3.2. Ultimately, Equation 25 can be applied

here to the three-dimensional channel with $n_W$ being the maximum value of $n$ for which $c_n > x + 2L + 2W$.

### 3.9   Baroclinic, closed channel, gradual forcing, rotating

The final step before incorporating melt is anticlimactic because the hard work has been done. To adapt

the closed, abrupt release case from the previous section to the gradual release case, we simply repeat the

methodology described by Equations (26) and (27).





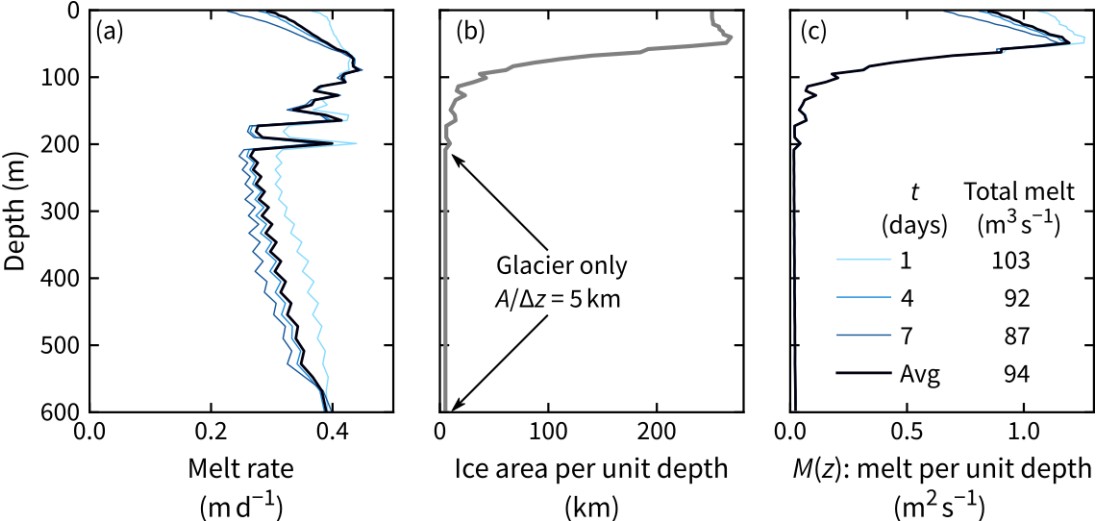

**Figure 9.** Iceberg (and glacier) melt in the numerical model. (a) Melt rates are ~0.3 m d$^{-1}$ and decrease over time, especially near the surface, as the surrounding water cools. (b) Total ice surface area in units equivalent to a width. (c) The product of panels a and b in units that are suitable for the analytical model.

### 3.10 Incorporating ice melt into the analytical model

In Sections 3.4–3.9, there is a low-density anomaly in the upper third of the water column in the $x < 0$ region that either exists at $t = 0$ or develops gradually as $t$ increases. The numerical model from Section 2 is set up similarly in that meltwater is continually injected into the $x < 0$ region.

In a glacier context, melt rates are often discussed in velocity units, with a convenient unit being m d$^{-1}$ as in Figure 9a. In iceberg-choked fjords, the total surface area of ice in contact with water can be an order of magnitude larger (Sulak et al., 2017). Indeed, expressing this surface area *per unit depth* gives a quantity that is equivalent to a glacier width, but much bigger, especially near the surface (e.g, ~200 km in Figure 9b).

Let $M(z)$ denote the total, time-averaged meltwater input per unit depth with units of m$^2$ s$^{-1}$. (The depth integral of $M(z)$ is the volume flux of meltwater.) Many remote sensing studies provide methods to estimate $M(z)$ from either observed rates of change of iceberg freeboard or parameterizations of melting (Enderlin and Hamilton, 2014; Enderlin et al., 2018; Moon et al., 2018; Moyer et al., 2019; Rezvanbehbahani et al.,



2020). We, however, have the benefit of the exact value of $M(z)$ being a model output, so we will use this

hereafter (Figure 9c).

To use the analytical model, we need an expression for $\rho_L(z)$ that plugs into Equation 27. We start with meltwater's properties (see, e.g., Jenkins, 1999):

$$S = 0 \tag{30}$$

$$\theta_{\text{eff}} \approx -\frac{L}{c} + \frac{c_i}{c}\theta_i \tag{31}$$

where $\theta_{\text{eff}}$ is the effective potential temperature of meltwater, $\theta_i$ is the ice temperature, $L$ is the latent heat of fusion, and $c_i$ and $c$ are the specific heat capacities of ice and water, respectively. The value of $\theta_{\text{eff}}$ is approximately $-85°C$ and is dominated by the first term, which accounts for the heat extracted from the water to induce the phase change; the second term accounts for sensible heating of the ice to its freezing point.

In Section 3.6, we defined the profile $\rho_L$ together with the time scale $\tau$ that defined how long it took for the left side to reduce from $\rho_R$ to $\rho_L$ at a constant rate. Here, we set a somewhat arbitrary value for $\tau$ of 7 days and calculate $\rho_L$ at this time based on the meltwater input. The value of $\tau$ later cancels when used in Equation 27.

At time $\tau$, the volume of meltwater per unit depth added over the mélange is $\Delta A(z) = M(z)\tau$. If the

fjord did not respond dynamically to this meltwater, then the average properties throughout the mélange region—based on weighted averages of the ambient water and meltwater—would become

$$S_L = \frac{A_0 S_R}{A_0 + \Delta A} \tag{32}$$

$$\theta_L = \frac{A_0 \theta_R + \Delta A \theta_{\text{eff}}}{A_0 + \Delta A} \tag{33}$$

$$\rho_L = \rho(S_L, \theta_L) \tag{34}$$

where $S_R$ and $\theta_R$ are the initial salinity and potential temperature profiles, and $A_0(z) = 2WL\phi(z)$ is the mélange area scaled by the water fraction $\phi(z)$. In our numerical model, we have $\phi = 0.9$ at the surface because 10% of the surface is covered by icebergs.





## 4    Comparing the analytical and numerical models

The analytical model should work in regions governed by linear wave dynamics; it will not work where

advection dominates. For example, at $t = 7$ days as in Figure 2, the analytical model will not work for $x \lesssim 15$ km. Hence, we will test it at $x = 20$ km by comparing its prediction to the numerical model from Section 2.

Consider first the cross-channel structure. At $t = 1$ day, the analytical model correctly predicts that the flow (i) has peak velocities of $1\,\mathrm{cm\,s^{-1}}$, (ii) has a zero-crossing at 100 m depth, and (iii) has a decay scale

comparable to the width of the channel (Figure 10a–b). Although it is a minor effect, the small patch of outflow centred near 350 m depth is also correctly predicted. Later, at $t = 3$ and 7 days (Figure 10c–f), there is a larger contribution from higher-modes at the $x = 20$ km mark, and the flows are consequently more concentrated in the top 100 m. The analytical model still predicts well the velocity fields at both of these times.

When compared carefully, the analytical model slightly overpredicts the velocities. This is best quantified by looking at the total outflow $Q_{\text{out}}$, which is the area integral of all outflowing fluid:

$$Q_{\text{out}} = \int_{u>0} u \, \mathrm{d}y \, \mathrm{d}z \qquad (35)$$

Again, we evaluate this at $x = 20$ km. Between 3 and 7 days (once a quasi-steady state is approached), the analytical model overpredicts $Q_{\text{out}}$ by 25% (compare the grey and black lines in Figure 11). However,

given the number of approximations and assumptions that go into the analytical model, we deem this a reasonable agreement. Indeed, one notable assumption not yet discussed is that the icebergs have no dynamical effect as obstacles. In reality and in the numerical model, icebergs induce drag on near-surface flows (Hughes, 2022), which includes the set of Kelvin waves that initially travel westward from $x = 0$ and then anti-clockwise around the fjord boundary.

To further test the analytical model, we repeated the $Q_{\text{out}}$ comparison for two other model scenarios. The first has stronger stratification: the salinity difference between the surface and seafloor is doubled ($\Delta S$ in Equation 1 is 6, not 3). The second has weaker melt rates: the turbulent transfer coefficients for heat and salt are four times smaller than the default case ($\gamma_T$ and $\gamma_S$ in Equations A3 and A4). In these two further





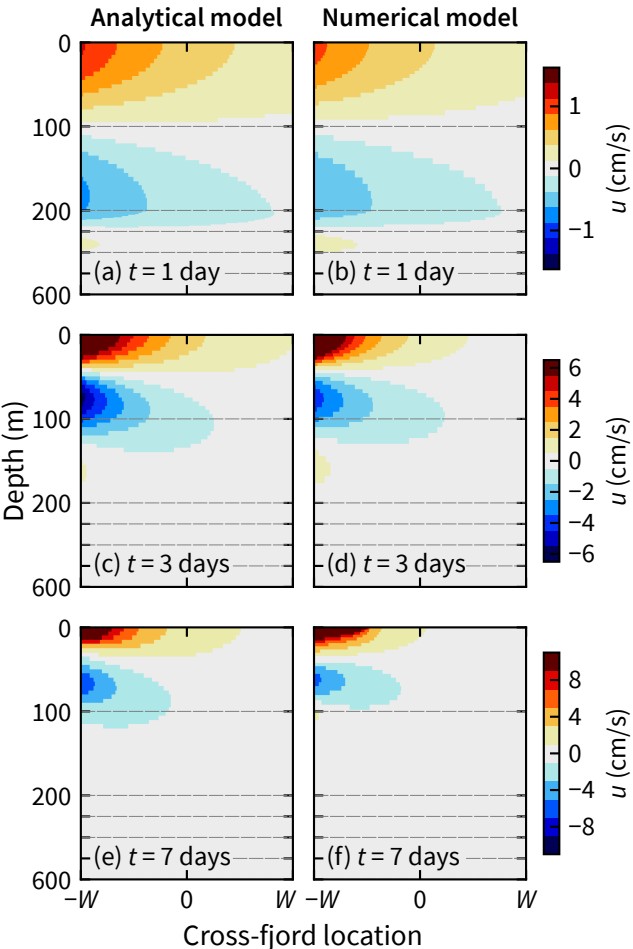

**Figure 10.** Snapshots at $x = 20\,\text{km}$ after 1, 3, and 7 days show that the analytical predictions in the left column agree with the numerically simulated velocities in the right column. Note that the color limits increase from the top to bottom and that the vertical axes are enlarged in the top 200 m as in Figure 3.

tests, the analytical model still predicts well the numerically simulated flow both in terms of total outflow
(Figure 11) and cross-channel structure (not shown).

Three comparisons is, of course, far from an exhaustive test of the plausible parameter space. Yet given the agreement in all cases, there is no reason not to trust the analytical model.





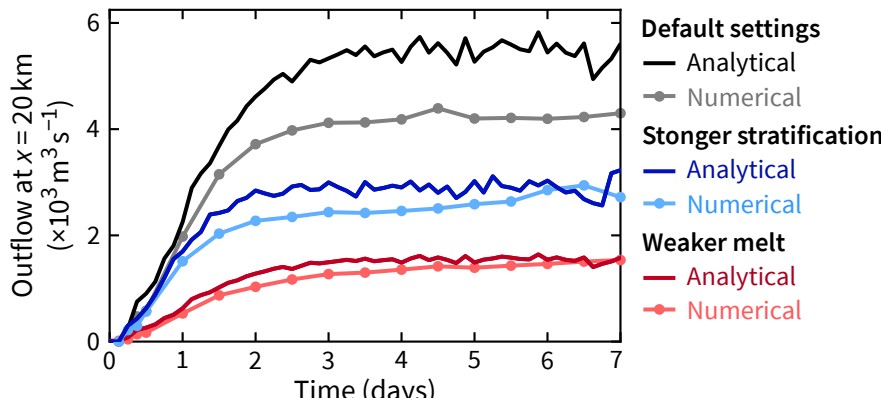

**Figure 11.** The analytical model predicts well the outflow (Equation 35) for three different scenarios. The slight overestimates are discussed in the main text.

## 5 Discussion

### 5.1 What parameters does melt-induced circulation depend on?

There are several obvious ways that melt-induced fjord circulation could increase: warmer ambient water, a larger mélange, or enhanced turbulent transfer at the ice–ocean interface.

To examine these dependencies in detail—and the less intuitive role of stratification—we undertake a parameter space study using the analytical model to predict $Q_{\mathrm{out}}(x = 20\,\mathrm{km})$ under a range of conditions. For simplicity, we will change one parameter at a time; all others will have the default values used

previously ($L = 8000\,\mathrm{m}$, $2W = 5000\,\mathrm{m}$, $H = 600\,\mathrm{m}$, $\Delta S = 3\,\mathrm{psu}$, and $\theta_{\mathrm{a}} = 2^\circ\mathrm{C}$). We will also assume a total melt rate profile similar to that in Figure 9c. Specifically,

$$M(z) = 2.5 \times 10^{-8}\, WL \left( 1 + \tanh\left( \frac{z + 100}{25} \right) \right) \tag{36}$$

where $100\,\mathrm{m}$ is the depth at which $M(z)$ drops to half of its surface maximum and $25\,\mathrm{m}$ is a vertical length scale. For the default fjord geometry, $M(z = 0) = 1\,\mathrm{m^2\,s^{-1}}$.

$Q_{\mathrm{out}}$ increases monotonically with channel width (Figure 12a). For narrower channels there is a dependence on $\sum_n Q_n \tanh(W/L_{\mathrm{R}n})$. This follows from generalizing Equation 6 to the baroclinic case involving a sum of modes, each with their own internal Rossby radius $L_{\mathrm{R}n}$. (Presumably the summation coefficients $Q_n$ could be derived from the analytical model, but that is not our goal here.) For wider channels $Q_{\mathrm{out}} \propto W + L$



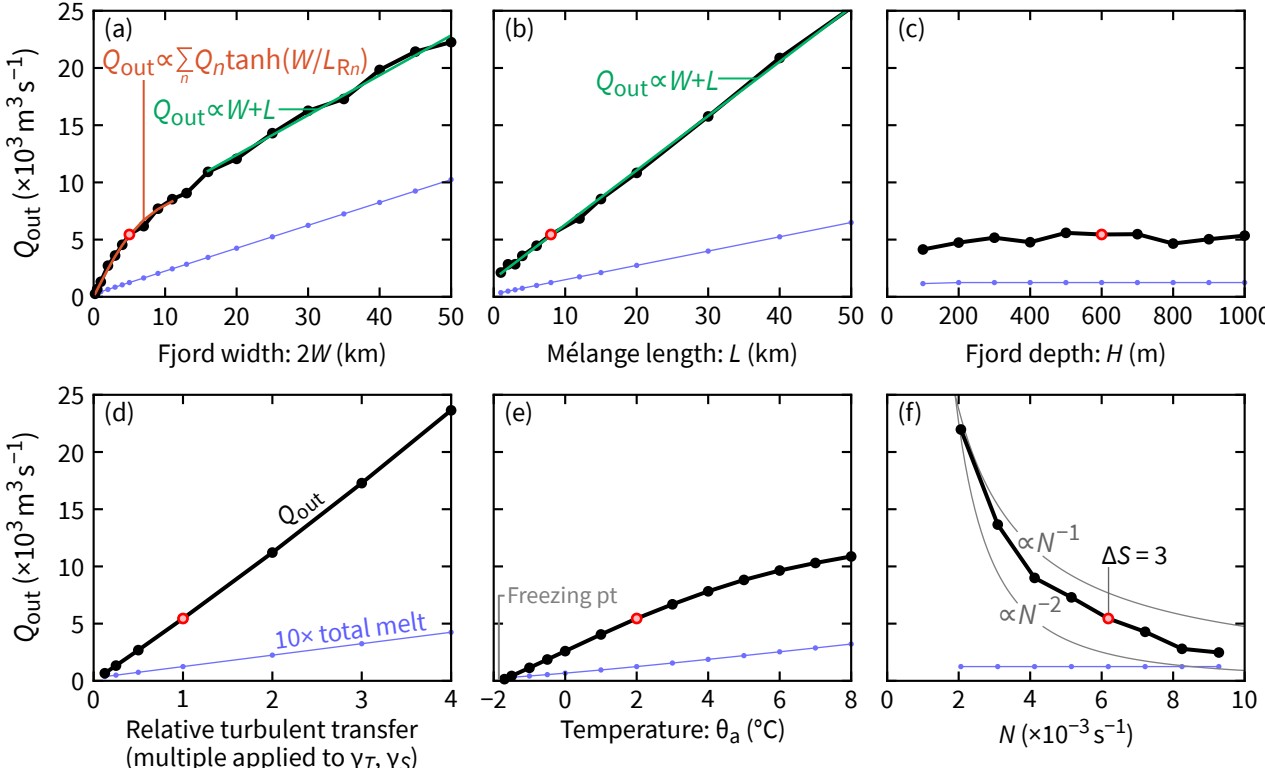

**Figure 12.** Predictions from the analytical model of how outflow ($Q_{out}$) depends on geometrical parameters, ice–ocean turbulent transfer, and water column properties. In each panel, single quantity is varied; the others are fixed at the values highlighted in the other five panels. Note how the total meltwater flux, which is shown for reference, is multiplied by a factor of ten to make it visible on the same scale.

because the distance $2L + 2W$ is the distance a Kelvin wave travels around the perimeter of the fjord to
move from $(0, +W)$ to $(0, -W)$, and this distance governs the total flux (see Sections 3.2 and 3.8). For the same reason, a $Q_{out} \propto W + L$ scaling also arises when varying $L$ while keeping $W$ constant (Figure 12b). The other geometrical parameter—fjord depth $H$—has no significant effect on $Q_{out}$ (Figure 12c).

    If the fjord geometry is fixed, but the ice–ocean interface conditions are changed, then $Q_{out}$ scales approximately linearly with the meltwater flux. In Figure 12d, we alter the ice–ocean turbulent transfer
coefficients $\gamma_T$ and $\gamma_S$ (Appendix A). In Figure 12e, we alter the ambient temperature. In both cases, linearity dominates; slight deviations from this arise from nonlinearities in the equation of state and the melt rate derived from the three-equation formulation. Note, however, that for this analysis we assumed a





fixed ambient velocity of $0.04\,\mathrm{m\,s^{-1}}$ (see Appendix A). Different melt formulations (e.g., Greisman, 1979; Magorrian and Wells, 2016; Malyarenko et al., 2020) may lead to different dependencies for $Q_{\mathrm{out}}$, but we

do not investigate these here.

$Q_{\mathrm{out}}$ vs water column stratification $N$ is the scaling needing the most steps to explain. First, recall the open channel, non-rotating system from Section 3.4. There, $u$ and hence $Q_{\mathrm{out}}$ are $\propto N^{-1}$ because $E_{\mathrm{p}} = E_{\mathrm{k}} \propto N^{-2}$ (Equation 21). However, $Q_{\mathrm{out}} \propto N^{-2}$ for the equivalent closed, gradual-release system (Section 3.6). The additional factor of $N^{-1}$ arises because internal wave speed $c \propto N$ (Equation 11).

Specifically, recall Figure 7 in which the gradual-release case was described as the sum of sequential versions of the abrupt-release case, with the latter consisting of $2L$-wide bands of nonzero velocity. These bands move and separate at a rate $\propto N$, and so the magnitude of their sum is $\propto N^{-1}$. This already elaborate explanation is further complicated by rotation because the internal Rossby radii $L_{\mathrm{R}n} = c_n/f \propto N/n$. With increasing $L_{\mathrm{R}n}$, flow occupies a wider portion of the channel and hence $Q_{\mathrm{out}}$ increases. Ultimately, in the

full system, $Q_{\mathrm{out}}$ has a scaling that tends to fall between $N^{-1}$ and $N^{-2}$ (Figure 12f).

All panels in Figure 12 include a line showing the total meltwater flux. In many cases, $Q_{\mathrm{out}}$ is $\sim$50 times larger than the meltwater flux, but in some cases the ratio is as large as 200 (small $L$ or small $N$) or as small as 20 (large $W$ or large $N$).

## 5.2   The value of simple scaling laws in Greenland fjords

We developed the above scalings to help tame the daunting problem of predicting the dynamics of fjords that are subject to numerous forcings, of which iceberg melt is only one. Of course, there is still much to do in extending our analytical model to one that is directly applicable to a realistic fjord. But we have taken the first steps to predicting the Greenland-wide role of iceberg melt.

An ambitious goal for our analytical model would be to make large-scale predictions like those that

exist for the role of subglacial discharge. These start with classic buoyant plume theory (Morton et al., 1956), extend it to a salinity-stratified system, and then apply it on a large scale. For example, Slater et al. (2016) show that outflow in a fjord is proportional to $Q_{\mathrm{sg}}^{3/4}/N^{5/8}$, where $Q_{\mathrm{sg}}$ is the subglacial discharge rate. Slater et al. (2022) applies this scaling to more than 100 fjords around Greenland to predict that a total of $20\,000\,\mathrm{m^3\,s^{-1}}$ of meltwater discharge gets amplified by a factor of $\sim$50 due to entrainment and that the

outflow is spread over the top $\sim$200 m.



A more immediate goal is to help interpret observations or numerical models for specific settings. For example, as part of ongoing related work, we are analyzing multi-year simulations of Sermilik Fjord with and without icebergs (using a setup like Davison et al. (2020), but with seasonal variability included). Iceberg effects are isolated by looking at the difference between the two simulations. One plausible but counter-intuitive hypothesis stemming from the analytical model is that melt-induced circulation will be larger in winter than in summer. Why? Because in winter the water column is less stratified (hence larger iceberg-melt-induced outflow) and this may overcome the effect of slower melting in cooler waters.

### 5.3 Next steps

The analytical model is built on linear wave dynamics; nonlinear advection and instabilities are ignored. In parts of the domain, this can quickly become a limitation. In the numerical model we saw advection dominating in the $0 < x < 10\,\mathrm{km}$ region after only one week of spin up (Figure 2a). In principle, it may be possible to extend the analytical model to predict this advective component. The approach would follow Hermann et al. (1989) from who we borrowed the analytical expressions for the barotropic wave-adjusted state in Section 3.1. For Hermann et al., the wave-adjusted state was merely the starting point for predicting the slower advective dynamics. In our *baroclinic* setting, however, the math would quickly become cumbersome.

Instead, it makes more sense to simply ask whether the analytical model would remain skillful after several months? Without running simulations to properly answer this, our best guess follows from Carroll et al. (2017) who simulated circulation induced by subglacial discharge on time scales of months. Their Figures 2 and 3 imply that spin up of the linear circulation (i.e., boundary currents) happens within the first five days. Thereafter, eddies form via instabilities and advection, especially in the wider channels. Nevertheless, their outflow metric seems mostly unaffected by these nonlinearities.

Other obvious uncertainties surround whether the analytical model still works in fjords that are not rectangles, whether it remains useful in the presence of competing forcings such as shelf waves and subglacial discharge, and how it should be adapted for the case where there is not a convenient demarcation of mélange and open water, but rather where iceberg concentration varies along the fjord.



## 6    Conclusion

The analytical model involved many steps. A summarizing example helps brings all of these steps together.

The continual input of meltwater generates a continual fjord response. Discretizing the problem in time
makes this response easier to understand. In Figure 13, we divide the problem into 100 pieces: 1% of the
meltwater is released at $t = 0.00t_0$, another 1% is released at $t = 0.01t_0$, and so on up to $t = 0.99t_0$. The
circulation at $t = t_0$ is the sum of these (Figure 13b).

For the response to the last 1% released at $t = 0.99t_0$, we see the lower modes move away quickly
from $x = 0$ with the higher modes trailing behind (Figure 13c). In front of the mode-1 wavefronts on
each side of the fjord, velocities are zero. Behind the mode-1 wavefronts, but in front of the mode-2
wavefronts, the velocities are down-fjord in the top half and up-fjord in the bottom half. These velocities
decay exponentially from the wall but, being mode-1, they extend a reasonable distance. Wavefronts for
the higher modes trail behind and their associated velocities decay rapidly with distance from the wall. The
same general velocity structure is present for the case with meltwater released at $t = 0.98t_0$, except that the
mode-1 wavefront on the far side has turned the corner (Figure 13d).

For the $t = 0.95t_0$ case (Figure 13e), the mode-1 wave that originated on the far side travels around the
fjord perimeter and starts to interfere with the velocity field generated by the other set of Kelvin waves.
The mode-2 wave does the same in the $t = 0.90t_0$ case (Figure 13f). With enough time, this interference
occurs for all modes. Indeed, for the $t = 0.00t_0$ case, the velocity field shown in Figure 13g is zero for
$x > 0$. This motionless region therefore has no influence on the total velocity field in Figure 13b.

Ultimately, Figure 13 implicitly illustrates that, down-fjord of $x = 0$, melt-induced fjord circulation
reaches a quasi-steady state in which it is only responding to the 'recent' input of meltwater, with 'recent'
linked to the time it takes for the relevant modes (say modes 1 through 10) to travel around the boundary.
For most most Greenland fjords, this will be only a few days.





**Figure 13.** Summary of the analytical model. To predict the total fjord circulation at a given time $t = t_0$ as in panel b, approximate the continual forcing as, say, 100 sequential, abrupt release cases, examples of which are shown in panels c–g. See Section 6 for a complete description. For clarity, velocities are shown only at the fjord edges and only the lowest four modes are considered. Color scales in panels c–g are the same, but these are different to that in panel b.





**Appendix A: Ice–ocean thermodynamics**

In our simulations, icebergs produce meltwater through only subsurface melting; wave erosion and melting
at the ice–air interfaces are ignored. Thermodynamics at all ice–ocean interfaces are treated with the
three-equation formulation, and the same velocity-dependent turbulent transfer coefficients are used for the
vertical sides and the basal face. Specifically, we adapt the 'icefront' package implementation from Xu
et al. (2012). Turbulent heat fluxes to the ice–ocean interface are

$$\text{heat transfer } [\text{W m}^{-2}] = \rho\, c_{\text{p}} \gamma_T |\mathbf{u}| \Delta T \tag{A1}$$

$$\text{salt transfer } [\text{psu m s}^{-1}] = \gamma_S |\mathbf{u}| \Delta S \tag{A2}$$

where $\Delta T$ is the difference between the temperature *at* the ice–ocean interface and the temperature in
the adjacent ocean cell, and similarly for $\Delta S$. The interface conditions come from the solution to the
three-equation formulation. The transfer coefficients for heat and salt in units of m s$^{-1}$ are $\gamma_T |\mathbf{u}|$ and $\gamma_S |\mathbf{u}|$
where

$$\gamma_T = 4.4 \times 10^{-3} \tag{A3}$$

$$\gamma_S = 1.24 \times 10^{-4} \tag{A4}$$

$$|\mathbf{u}| = \min\left(\sqrt{u^2 + v^2 + w^2}, 0.04\,\text{m s}^{-1}\right) \tag{A5}$$

The values of $\gamma_T$ and $\gamma_S$ are far from well constrained; the decimal places are shown only to help preserve a
link to previous studies that used $\gamma_T = 1.1 \times 10^{-3}$ and $\gamma_S = 3.1 \times 10^{-5}$ (e.g., Xu et al., 2012; Sciascia et al.,
2013). Observations suggest these values are too low at vertical or near-vertical ice faces in Greenland
(Jackson et al., 2020; Schulz et al., 2022). We have increased $\gamma_T$ and $\gamma_S$ by a factor of four following the
suggestion of Jackson et al. (2020, 2022)[1] based on their scenario in which a resolved horizontal velocity
is incorporated into the calculation of the transfer coefficients (rather than just vertical velocity). In our
case, $|\mathbf{u}|$ values are calculated in the cells adjacent to ice–water interfaces. At each interface, two of the
three velocity components will be nonzero. For example, $v \neq 0$ and $w \neq 0$ for an ice face in the $y$–$z$ plane.
The 0.04 m s$^{-1}$ lower limit follows Slater et al. (2015) and is intended to represent unresolved melt-driven
convection.

---

[1]Jackson et al. (2020) and some other studies define the conventional values of $\gamma_T$ and $\gamma_S$ in an alternate way. Namely,
$\gamma_T = \sqrt{C_{\text{d}}}\Gamma_T$ and $\gamma_S = \sqrt{C_{\text{d}}}\Gamma_S$ where $C_{\text{d}} = 2.5 \times 10^{-3}$, $\Gamma_T = 2.2 \times 10^{-2}$, and $\Gamma_S = 6.2 \times 10^{-4}$.



## Appendix B: Extension to arbitrary stratification

Conceptually, the analytically model does not change if the reference density is nonlinear. The only difference is that mode shapes and internal wave speeds need to be calculated numerically with matrix methods.

Mode shapes for vertical velocity, denoted $\phi_n^w$, are the eigenvectors of the following equation:

$$\frac{\partial^2}{\partial z^2}\phi_n^w + \frac{N^2}{c_n^2}\phi_n^w = 0 \tag{B1}$$

with boundary conditions requiring that $\phi_n^w = 0$ at the surface and seafloor. The internal wave speeds $c_n$ are derived from the eigenvalues. We are interested in the horizontal velocity mode shapes, which we denote $\phi_n$ without a superscript:

$$\phi_n = \frac{\partial \phi_n^w}{\partial z} \tag{B2}$$

It is easy to confirm that $\phi_n = \cos(n\pi z/H)$ and $c_n = NH/n\pi$ are eigensolutions to the above set of equations if $N$ is constant. Extending the analytical model to nonlinear stratification simply involves replacing all appearances of $\cos(n\pi z/H)$ and $NH/n\pi$ with $\phi_n$ and $c_n$, respectively.

*Code availability.* The archive at doi.org/10.5281/zenodo.8339482 includes (i) the analytical model written in Python, (ii) all the code and configuration files necessary to recreate the MITgcm results, and (iii) snapshots of these results in netCDF format. The analytical model is also available at github.com/hugke729/IcebergMeltCirculation.

*Competing interests.* The author declares that they have no conflict of interest.

*Financial support.* This work was funded by the Heising-Simons Foundation (grant #2019-1159) as part of the program 'Eyes at the front: a megasite project at Helheim Glacier.' The numerical simulations were run on the Expanse system at the San Diego Supercomputer Center through allocation EES220032 from the Advanced Cyberinfrastructure Coordination Ecosystem: Services and Support (ACCESS) program, which is supported by National Science Foundation grants #2138259, #2138286, #2138307, #2137603, and #2138296.



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
