# Peer review of "Fjord circulation induced by melting icebergs"

_EGUsphere, 2023_

## Author Comment (AC1)

**Fjord circulation induced by melting icebergs**

Kenneth G. Hughes[1]

[1]College of Earth, Ocean, and Atmospheric Sciences, Oregon State University, Oregon, USA

**Correspondence:** Kenneth Hughes (kenneth.hughes@oregonstate.edu)

**Abstract.** TEXT

*Copyright statement.* TEXT

**1 Introduction**

Rivers of meltwater discharge from the base of glaciers in Greenland at rates of tens to hundreds of cubic
5 meters per second. This fresh, buoyant water rises as a plume hundreds of meters up the face of the
glacier, entraining many times more ambient seawater as it goes. If the discharge is high or the water
column stratification weak, the plume reaches the surface and transitions to a down-fjord, surface current.
Otherwise, the plume reaches neutral buoyancy and flows out at mid-depth. Either way, typical outflow
speeds are $\mathcal{O}(10)\,\mathrm{cm\,s^{-1}}$ (e.g., Bendtsen et al., 2015; Jackson et al., 2022). This induces thicker, slower
10 flows at depth to compensate. Over a summer, this plume-driven circulation is often enough to renew
waters in the fjord (e.g., Gladish et al., 2015; Slater et al., 2022). Glacier melt can also induce circulation,
but it is an order of magnitude weaker (Sciascia et al., 2013; Cowton et al., 2015).

If subglacial discharge is known, other quantities can be predicted. Slater et al. (2016), for example,
uses buoyant plume theory to show that for a given subglacial discharge rate $Q_{\mathrm{sg}}$ in a salinity-stratified
15 system with buoyancy frequency $N$, plume volume flux (and hence outflow) is proportional as $Q_{\mathrm{sg}}^{3/4}/N^{5/8}$
(see their Eq. 15). Glacier melt induced by a single plume scales as $Q_{\mathrm{sg}}^{1/3}$ or $Q_{\mathrm{sg}}^{2/5}$ depending on the details
(Jenkins, 2011; Cowton et al., 2015). A corollary of these sublinear dependencies on $Q_{\mathrm{sg}}$ is that the same
total discharge will lead to larger total volume fluxes and melting if the discharge is distributed among

multiple plumes (Slater et al., 2015; Carroll et al., 2015) or over a longer melt season (Cowton et al., 2015).

These simple scalings provide the building blocks for more elaborate theories or wider-ranging applications. Zhao et al. (2021) use subglacial discharge to predict well the magnitude of overturning and horizontal recirculation in idealized fjords across a range of fjord widths, sill heights, wind forcing, and stratifications. And Slater et al. (2022) applies plume models to more than 100 fjords around Greenland to estimate that a total of $20\,000\,\mathrm{m^3\,s^{-1}}$ of meltwater discharge gets amplified by a factor of ~50 due to entrainment. That is, a freshwater flux that is initially small by oceanic standards gets amplified to something that is better described in the oceanographer's unit of Sverdrups.

Subglacial discharge is not the only significant source of cool freshwater in many Greenland fjords. Iceberg melt is another and 'nowhere in the sea could a melting iceberg be expected to have a more pronounced effect on its environment than in the enclosure of a fjord' (Gade, 1979). At any given time, Sermilik Fjord is home to $\mathscr{O}(10\,000)$ icebergs and, as a result, the annual-average freshwater flux from melting icebergs is comparable to or larger than subglacial discharge (Moon et al., 2018; Moyer et al., 2019; Rezvanbehbahani et al., 2020). But the influence of this distributed meltwater source is not clear. As Enderlin et al. (2016) note, 'studies of fjord circulation and feedbacks [...] have largely ignored the contribution of iceberg melt within the fjords as a driver of fjord circulation'.

There are currently two complementary lines of study that shed light on the effects that icebergs and their melt have on fjord dynamics: (i) detailed studies of flow and melt of a single iceberg and (ii) fjord-scale simulations with icebergs as subgrid-scale components.

Dye releases and surveys near 100-m scale icebergs show upwelling at $7\,\mathrm{cm\,s^{-1}}$ beside the ice and horizontal extents of meltwater traceable 500–1000 m away (Josberger and Neshyba, 1980; Yankovsky and Yashayaev, 2014). Laboratory experiments of 10-cm scale icebergs in horizontal flows show that increasing the current increases the melt rates if the flow is $\gtrsim 2\,\mathrm{cm\,s^{-1}}$; below this, melt rate is insensitive to speed (FitzMaurice et al., 2017; Hester et al., 2021). Together, these results imply that, in a fjord with many icebergs, the meltwater from individual icebergs will almost certainly merge and that this combined meltwater flux could have a positive feedback as alluded to earlier: increased melt leads to increased velocities that further increases the melt rate given its velocity dependence.

Need to be careful with whether icebergs melting in presence of stratification. I think Josberger at least are in unstratified water

Davison et al. (2020) developed a subgrid-scale iceberg parameterization and implemented into simulations of Sermilik Fjord with $500{\times}500$ m horizontal resolution. Their simulations incorporated $\sim$10 000 cuboid icebergs distributed among 1400 cells with many of these cells ($\sim$40%) housing a single iceberg and others ($\sim$10%) housing more than 10 icebergs. Melt of these icebergs—inferred from each grid cell's temperature, salinity, and depth—lead to a meltwater flux distributed in three dimensions. In absence of subglacial discharge, but an otherwise summertime scenario with surface waters at 4°C, they found that iceberg melt induces slow currents ($\lesssim$0.02 m s$^{-1}$) over the top 200 m. Since this water did not flush through the system quickly, the cooling and freshening effects of icebergs were strong: up to 5°C and 0.7 psu. Adding in a large subglacial discharge of 1200 m$^3$ s$^{-1}$ increased flushing and thereby reduced these effects by a factor of five. Using the same iceberg parameterization in a simulation of Ilulissat Icefjord, **?** found that these cooling and freshening effects improved agreement between model and observation.

Here, we examine the roles of icebergs—thermodynamical and mechanical—on fjord dynamics by developing simulations akin to that of Davison et al. (2020) and **?**, but at 30–50$\times$ their horizontal resolution (albeit over shorter time and space scales). This enables us to zoom in and explicitly resolve flow in, around, and under icebergs. In essence, our aim is to combine facets of the detailed, single iceberg studies and the fjord-scale studies. This aim grew out of a prequel study (Hughes, 2022) highlighting the nontrivial behavior of stratified flow when it impinges on and traverses a field of icebergs, rather than a single iceberg. That study, however, lacked iceberg thermodynamics, a glacier, and a subglacial discharge plume. We now add these processes.

**2 Model setup**

Our model setup builds on that of Hughes (2022), which had a melange comprising thermodynamically-inactive icebergs in the center of an open channel and was forced by imposed velocities at the boundaries. Here, we increase the realism by adding a glacier as a boundary at one end of a fjord and using meltwater

[Figure]

**Figure 1.** Model setup. In existing setup, I messed up and ended up with a glacier face at the far right end as well.

as the agent driving circulation, which is input through both melt at the ice–ocean interfaces and subglacial discharge.

75     Simulations are undertaken with the MITgcm (Marshall et al., 1997; Adcroft et al., 2004), in which it is possible to add volume-occupying ice cells at the surface (Losch, 2008) to serve as the icebergs.

**2.1   Model domain**

All simulations have a 3.8 km wide and $H = 600$ m deep fjord closed at one end by a glacier. The domain extends 100 km in the along-fjord (east–west) direction, but we focus on the first 4.5 km, which contains 80   the melange (Figure 1a). Within this region, $\Delta x = \Delta y = 10$ m. Beyond this, $\Delta x$ increases 3% per cell. There are 64 vertical levels, with the highest resolution of $\Delta z = 3$ m at the surface.

A time step of $\Delta t = 1\,\mathrm{s}$ (sometimes $1.5\,\mathrm{s}$) is used and simulations are run for 56 hours to allow a quasi steady state to develop for most cases (Appendix A). Unless otherwise noted, all results shown will be snapshots at the end of a simulation. Given this short simulation time (and lack of a sill), we are not addressing issues such as seasonality, deep basin circulation, or intermediary circulation (several-day-long baroclinic fluctuations driven at the fjord mouth).

For most simulations, the ambient water initially has a constant temperature and linear salinity stratification:

$$T_a(z) = 2°C \tag{1}$$

$$S_a(z) = 34.5 - 0.5z/H \tag{2}$$

Typical changes in these quantities by the end of the simulation are 0.02–0.1°C and 0.1–0.3 psu (Appendix A).

For cold waters like these, salinity has the dominant role on density; temperature then acts as a tracer (albeit not passive nor conserved). The 2°C value equates to approximately 4°C above the freezing, an average thermal forcing for Greenland fjords (Wood et al., 2021). The salinity increase $\Delta S = 0.5$ between the surface to the seafloor is lower than the more typical values of 2–4 observed in Greenland fjords (e.g., Straneo et al., 2011; Sciascia et al., 2013). We use the weak stratification and constant temperature for pedagogical reasons. Weak stratification lets the subglacial discharge plumes reach the surface, which induces a simple circulation regime: a single overturning cell. Constant temperature means that the effect of melting is always to cool water. If temperature varied with depth, then vertical advection induced by melting can lead to local warming at mid depths (Davison et al., 2022), thereby complicating our analysis. Sensitivity studies will use stronger salinity stratification or observed temperature and salinity profiles for initialization.

De Andrés et al. (2020) talks at length about role of stratification on plume neutral buoyancy

For $\Delta S = 0.5$, the mode-1 wave speed is approximately $c = 0.5\,\mathrm{m\,s^{-1}}$. With the Coriolis frequency set to $f = 1.37 \times 10^{-4}\,\mathrm{s^{-1}}$ (70°N), the internal Rossby radius $c/f$ is approximately the same as the width of the fjord. If $\Delta S$ increases, so does the width of the fjord relative to the internal Rossby radius.

Will probably change 56 h to 48 h to be round number of 2 days. Most runs actually end at 32 h.

I'm going to redo all the runs on a larger domain. Will probably make the melange more like 7 or 8 km long and 5 km wide. I'll also play around with using shapes other than cuboids. Might also make default discharge 200 rather than $100 \, \mathrm{m^3 \, s^{-1}}$

**2.2   Melting icebergs**

The icebergs making up the melange are cuboids with horizontal dimensions of multiples of 40 m (Figure 1c). Therefore, flow around individual icebergs is explicitly resolved, albeit coarsely. The size distribution of these icebergs follows from the power law given by (Sulak et al., 2017) for observations from Sermilik Fjord: the number of icebergs of a given horizontal area goes as $A^{-1.9}$. That is, smaller icebergs are more numerous than larger icebergs. Although larger ones often occur in reality, we impose a maximum iceberg size of 320×240 m. Most icebergs have a keel depth of 30–80 m. Further details and justification of the fixed cuboid approach are given by Hughes (2022).

The only way that the melange varies between simulations is in the number of icebergs within the region. We denote the fractional area of the sea surface covered by icebergs as $\lambda$. The example in Figure 1a has $\lambda = 0.1$, meaning that 10% of the sea surface is ice. At $x = 4.5 \, \mathrm{km}$, this percentage drops abruptly to zero. This means that the melange has a sharp edge and helps us distinguish between the melange and non-melange regions.

Icebergs in our simulations input freshwater through subsurface melting but not through wave erosion or melting at the ice–air interface. Thermodynamics at all ice–ocean interfaces are treated with the three-equation formulation, and the same velocity-dependent turbulent transfer coefficients are used for the vertical sides and the basal face. Specifically, we adapt the ICEFRONT package implementation from Xu et al. (2012). Turbulent heat fluxes to the ice–ocean interface are

$$\text{heat transfer} \, [\mathrm{W \, m^{-2}}] = \rho c_\mathrm{p} \gamma_T |\mathbf{u}| \Delta T \tag{3}$$

$$\text{salt transfer} \, [\mathrm{m \, s^{-1}}] = \gamma_S |\mathbf{u}| \Delta S \tag{4}$$

where $\Delta T$ and $\Delta S$ are the difference between (i) the temperature and salinity in the ocean cells adjacent to ice and (ii) those at the ice–ocean interface. The transfer coefficients for heat and salt in units of $\mathrm{m \, s^{-1}}$

are $\gamma_T|\mathbf{u}|$ and $\gamma_S|\mathbf{u}|$ where

$$\gamma_T = 4.4 \times 10^{-3}, \tag{5}$$

$$\gamma_S = 1.24 \times 10^{-4}, \text{ and} \tag{6}$$

$$|\mathbf{u}| = \min\left(\sqrt{u^2 + v^2 + w^2}, 0.04\,\mathrm{m\,s^{-1}}\right) \tag{7}$$

The values of $\gamma_T$ and $\gamma_S$ are far from well constrained; the significant figures are merely to help maintain a link to previous studies that used $\gamma_T = 1.1 \times 10^{-3}$ and $\gamma_S = 3.1 \times 10^{-5}$ (e.g., Xu et al., 2012; Sciascia et al., 2013). Observations suggest these values are too low at vertical or near-vertical ice faces in Greenland (Jackson et al., 2020; Schulz et al., 2022). We have increased $\gamma_T$ and $\gamma_S$ by a factor of four following the suggestion of Jackson et al. (2020, 2022) based on their scenario in which a resolved horizontal velocity is incorporated into the calculation of the transfer coefficients (rather than just vertical velocity). In our case, $|\mathbf{u}|$ values are calculated in the cells adjacent to ice–water interfaces. At each interface, two of the three velocity components will be nonzero; for example, $v \neq 0$ and $w \neq 0$ for an ice face in the $y$–$z$ plane. The $0.04\,\mathrm{m\,s^{-1}}$ lower limit follows Slater et al. (2015) and is intending to represent unresolved melt-driven convection.

(Jackson et al. (2020) and some other studies define the conventional values of $\gamma_T$ and $\gamma_S$ in an alternate way. Namely, $\gamma_T = \sqrt{C_d}\Gamma_T$ and $\gamma_S = \sqrt{C_d}\Gamma_S$ where $C_d = 2.5 \times 10^{-3}$, $\Gamma_T = 2.2 \times 10^{-2}$, and $\Gamma_S = 6.2 \times 10^{-4}$.)

I haven't yet increased default values by factor of four

Need to set transfer coefficients within plume

Do sensitivity study with Schulz parameterization?

**2.3 Subglacial plume forcing**

A single subglacial discharge plume issues from the grounding line in the center of the glacier (actually slightly off center to avoid a numerical instability.) Its dynamics are calculated using buoyant plume theory, which comprises a set of coupled ordinary differential equations in the vertical direction. At every time step, a steady state plume solution is found using the current temperature and salinity at the plume's location. This approach is typically treated in the MITgcm as a subgrid-scale parameterization (the ICE-PLUME package; Cowton et al., 2015).

Need to move plume away from exact center

Here, a subglacial plume does not fit in a single cell, so we use the adapted approach of Zhao et al. (2022) in which the plume's entrainment and outflow is spread horizontally over a semi-circle (Figure 1b). Specifically, the extra mass flux is distributed over a 10-cell (100 m) radius, but weighted toward the center such that half the mass is added within a 5-cell radius. These horizontal scales roughly agree with the plumes observed at the surface at, say, Helheim Glacier (Melton et al., 2022). Vertically, outflow of the extra mass of fresher water is spread over five cells (or three or four if the plume reaches or nears the surface). Given the $\Delta z$ is variable, the initial thickness of the outflowing layer depends on the plume's depth of neutral buoyancy. It will typically be 10–30 m, which is thin but comparable to the limited observations of near-field plume outflow (e.g., Jackson et al., 2017).

With 10 m horizontal resolution it is arguably possible to explicitly simulate a plume but this is computationally expensive in that it requires the model to be run nonhydrostatically, and the output is sensitive to the choice of viscosity and diffusivity (Xu et al., 2012; Sciascia et al., 2013; Slater et al., 2015; Gladish et al., 2015; De Andrés et al., 2018, e.g.). Since our focus is on the larger fjord dynamics, not the plume itself, we can avoid these challenges. We recognize, however, that we lose the ability to have a plume that overshoots its neutral buoyancy depth and then oscillates over the next ∼500 m as it travels down-fjord (Carroll et al., 2015). And no ability to generate internal waves? Cite Jesse's paper if it is accepted at some point.

Cowton et al. (2015) uses entrainment coefficient of $\alpha = 0.1$, but this gives a dilution rate of O(100), which seems to large. Need to look into this before rerunning models. Then again, Gladish et al. (2015) got amplification factor of 280...

I'm currently putting in plume at a temperature of 0C. Should really be using pressure-dependent freezing point, but this is pretty similar (-0.45C for freshwater at 600m)

**2.4  Other model details**

Vertical mixing is parameterized with the Klymak and Legg (2010) overturning scheme with a background viscosity and diffusivity of $A_v = K_v = 10^{-4}\,\mathrm{m^2\,s^{-1}}$. Horizontal viscosity is parameterized with a Smagorinksy viscosity with the `viscC2Smag` coefficient set to 2.5 together with a background value of

$A_{\mathrm{h}} = 10^{-3}\,\mathrm{m^2\,s^{-1}}$. Temperature and salinity are advected with a third-order flux limiter scheme (termed scheme 33 in the MITgcm).

190    Slater et al. (2018) and Hager et al. (2022) use 2.2 for Smagorinksy. Does this difference matter?
    Carroll et al. (2015) and Sciascia et al. (2013) use vertical viscosity of $10^{-3}\,\mathrm{m^2\,s^{-1}}$. But maybe I should use something smaller?

All model runs are hydrostatic, which are approximately four times faster than nonhydrostatic simulations for our set up. Need to do proper comparison between modes.

**195  3   Results**

**3.1   Melt-driven circulation**

Iceberg and glacier melt, by themselves, drive a weak overturning circulation (Figure 2). Maximum velocities of a few $\mathrm{cm\,s^{-1}}$ occur at the surface as the input of buoyant meltwater creates an along-fjord pressure gradient. Under the influence of Coriolis, and downstream of the melange, the fastest flows occur on the
200    southern side of the fjord: the beginning of a coastal current appears at $x = 6\,\mathrm{km}$ in Figure 2a.

Within the melange, the fastest flows are on the northern side of the fjord. To understand this, consider a number of stationary icebergs melting into an unbounded region. In this case, meltwater would initially flow out in all directions and then, under Coriolis, evolve into clockwise eddy. This process would play out in manner similar to the buoyant meltwater plume next to a single iceberg (Yankovsky and Yashayaev,
205    2014) or, more generally, the Rossby adjustment of isolated lenses as described by Stuart et al. (2011). This tendency toward clockwise rotation is still evident in Figure 2b—note, for example, the surface inflow on the southern side—albeit modulated by the boundaries. (The Rossby adjustment of a cross-channel discontinuity in an open channel (Hermann et al., 1989) is instructive here. In that situation an asymmetric pattern arises with an eastward boundary current on the northern wall turning right—with
210    some overshoot—and crossing the channel to become an boundary current on the southern wall. Much the same occurs here if we consider the end of the melange as the discontinuity.)

Meltwater input is significant at depths shallower than $100\,\mathrm{m}$ (Figure 2i), but not all of this flows outward. Instead, there is a net inflow below $50\,\mathrm{m}$ depth (Figure 2e). This inflow is necessary to make

up for the divergence of vertical velocity at 50 m (Figure 3f), which is primarily caused by the depth-dependent input of meltwater (Figure 3g).

In the example in Figure 2, $\lambda = 0.1$ and there is $8 \times 10^6 \, \text{m}^2$ of ice in contact with water with a total meltwater input of $12 \, \text{m}^3 \, \text{s}^{-1}$. By comparison, Moon et al. (2018) estimate that in the first 20 km of Sermilik Fjord (the melange region), the input of iceberg melt is 60 and $200 \, \text{m}^3 \, \text{s}^{-1}$ for winter and summer, respectively. Hence, when scaled by a factor of approximately 4 to account for the shorter melange in our simulation, $12 \, \text{m}^3 \, \text{s}^{-1}$ is comparable to Moon et al.'s winter estimate.

Could also compare against latent heat of melting of $10^{12}$ W from Jackson and Straneo (2016)

**3.2 Plume-driven circulation**

Plume-driven flows are faster than iceberg-melt driven flows. For the example in Figure 3 with $Q_{\text{sg}} = 100 \, \text{m}^3 \, \text{s}^{-1}$, surface currents away from the plume are $\mathcal{O}(10) \, \text{cm} \, \text{s}^{-1}$ and have sufficient inertia to travel $\mathcal{O}(10) \, \text{km}$ before settling into a narrow coastal current. Closer to the glacier, an eddy forms. The structure of such eddies is sensitive to the fjord geometry and location of plume outflow (e.g., Carroll et al., 2017; Slater et al., 2018).

This plume-only run has a freshwater flux that is six times larger than the previously described iceberg-only run. However, this flux is flushed through the melange region ($x < 4.5 \, \text{km}$) more quickly and, consequently, the salinity deficit is only two times larger (compare Figs 2h and 3h). The temperature deficit is smaller in the plume run because adding cold water is less effective at cooling compared to extracting heat by melting ice. (Improve this explanation.)

**3.3 How icebergs influence plume flow**

Icebergs alter plume outflow in two clear ways: few larger eddies are replaced by many smaller eddies, and outflow is redistributed with depth.

Flow trajectories from the plume-only simulation in the previous section are shown in Figure 4 together with a simulation with both a plume and icebergs ($Q_{\text{sg}} = 100 \, \text{m}^3 \, \text{s}^{-1}$ and $\lambda = 0.1$). The large eddy on the southern wall centered near $x = 3 \, \text{km}$ does not arise when icebergs are present. This effect holds even when iceberg density is as low as $\lambda = 0.02$ (not shown). Evidently, the icebergs disrupt/introduce

[Figure]

**Figure 2.** Dynamics driven only by ice meltwater as depicted in (a, b) plan view, (c) an along-fjord slice, and (d) a cross-fjord slice. The meltwater's tendency to circulate clockwise at the surface manifests as the stronger eastward velocities at the north end. Away from the melange, a buoyant coastal current forms, the start of which is visible at $x = 6$ km. At 50–200 m depth, mean flow within the melange region is westward (panel e). This inflow counteracts the divergence of the vertical flow (panel f), much of which is caused by rising meltwater. Cooling and freshening of water within the melange is concentrated in the top 100 m (panels g and h). This example has $\lambda = 0.1$. Add panel i label. Note that only top half is shown.

[Figure]

**Figure 3.** Dynamics driven by a subglacial discharge plume. (The only melting ice is the glacier). A surface jet and slow return flow (panel e) are caused by the average upward flow (panel f), which is dominated by the plume's vertical velocities of $\sim 1\,\mathrm{m\,s^{-1}}$. Cooler, fresher water is concentrated in the top 50 m. Note the different color scale and $x$ limits compared to Figure 2. This example has $Q_{\mathrm{sg}} = 100\,\mathrm{m^3\,s^{-1}}$. Add panel h label. Match T and S limits to previous figure.

240  incoherency/something else... Figure out why and flesh out this argument. Instead of this large eddy, we see smaller ($\sim 100\,\mathrm{m}$-scale) eddies behind most icebergs.

[Figure]

**Figure 4.** Surface circulation without and with icebergs. (a–b) Without icebergs, near-plume trajectories are in all directions; an eddy forms 2–4 km from the plume; and a coastal current forms beyond that. (c–d) With icebergs, the far-field circulation is similar, but the near-field circulation is ... The circulation shown is based on offline advection of the 760 evenly spaced particles for 24 hours based on the final (quasi steady) velocity field.

Each iceberg acts as a drag element and, together, they slow the plume outflow at the surface. For $Q_{sg} = 100\,\mathrm{m}^3\,\mathrm{s}^{-1}$ and without icebergs, the mean current speed in the top 20 m over $x < 4.5\,\mathrm{km}$ is $15\,\mathrm{cm}\,\mathrm{s}^{-1}$ and the 95th percentile is $45\,\mathrm{cm}\,\mathrm{s}^{-1}$ (Figure 5). These speeds are approximately halved for $\lambda = 0.1$. Although the presence of icebergs introduces constrictions—which accelerate flow within—the maximum surface currents are, at best, only as fast as the average speed in the iceberg-free case.

Despite influencing current speeds, the number of icebergs has minimal influence on the overall outflow. We define $Q_{out}$ as the cross-sectional flux of outflowing fluid:

$$Q_{out}(x) = \int\int u(x,y,z)|_{u>0}\,\mathrm{d}y\,\mathrm{d}z \tag{8}$$

[Figure]

**Figure 5.** Near-surface flow velocities decrease as the number of icebergs increases. The velocity distribution comes from all grid cells within the melange region ($x < 4.5$ km) in the top 20 m. Distributions are offset vertically for clarity. Add xtick marks for each baseline. Decide whether to include $\lambda = 0.2$, or note why not.

250     The net flux is $Q_{sg}$, but this net is the small difference of two much larger terms. For $Q_{sg} = 100 \, \mathrm{m^3 \, s^{-1}}$, $Q_{out}$ is two orders of magnitude larger (Figure 6c). Much of this increase happens as the plume is rising vertically and entraining ambient water (a process that is parameterized here with a one-dimensional plume model; Section 2.3). Entrainment continues, albeit more slowly, as the outflow moves down fjord and mixes vertically.

255     Since surface flows are reduced with increasing $\lambda$, but $Q_{out}$ is comparatively constant, mid-depth flows must increase with $\lambda$. Within the melange, the fastest flows are not found at the surface once $\lambda$ exceeds approximately 0.1 (Figure 6a). For $\lambda = 0.2$ and 0.3, outflow peaks at 60–70 m where there are fewer icebergs to impede motion. (Only a third of the icebergs in our setup extend down to these depths.) However, when the outflow passes the end of the melange, it becomes surface bound again regardless of 260     its behavior beneath the melange (Figure 6b). To first order, the velocity structures are the same (including the cross-sectional structure; not shown).

[Figure]

**Figure 6.** Among simulations with differing iceberg concentrations $\lambda$, differences in average cross-fjord velocity are (a) large within the melange but (b) small beyond the melange. (c) Regardless of location, overturning strength (Eq ?) is largely independent of $\lambda$. Only the top third of the domain is shown in panels a and b. These simulations all have $Q_{\text{sg}} = 100\,\text{m}^3\,\text{s}^{-1}$. Check calculation of overturning strength within melange

Downstream of the melange, near-surface flow is easy to characterize as a geostrophic balance. The Coriolis and pressure gradient forces, both of which are large and nearly perpendicular to the fjord axis, cancel out (Figure 7).

265 One of Carroll et al. (2017)'s results was showing geostrophically balanced recirculation cells downstream of the glacier. Fraser et al. (2018) also noted that circulation is in geostrophic balance to a close approximation.

Within the melange, the pressure gradient and Coriolis terms are still the largest, but they are misaligned. The pressure gradient force—dominated by the sea surface height gradient $\nabla\eta$—has a significant

270 component in the down-fjord direction. For $\lambda = 0.2$ as in Figure 7, $-\nabla\eta$ is $2\,\text{mm}\,\text{km}^{-1}$ and is directed toward an angle of 50° from north. For simulations with more icebergs $-\nabla\eta$ gets larger and turns further

[Figure]

**Figure 7.** Large differences in the momentum budget inside and outside the melange for $\lambda = 0.2$. All quantities shown are averages over the top 20 m. Momentum budget terms are further averaged within the two dashed boxes.

down-fjord, and vice versa. With $\lambda = 0.05$, for example, $-\nabla\eta$ is $1\,\mathrm{mm\,km^{-1}}$ at an angle of $15°$ from north. More generally, $|\nabla\eta| \sim \lambda^{0.5}$.

The near-surface flow is not moving perpendicular to this gradient as would occur in geostrophic bal-
275 ance. Instead, its direction is closer to east-northeastward and consequently the Coriolis term has a positive component in the $x$ direction.

Acting against these forces is viscous dissipation. In terms of being a momentum sink, the dissipative term is significant where there is vorticity injected as boundary layers separate. This occurs in the vicinity of each iceberg, and so within the melange, dissipation plays a first-order rule in the momentum budget.

280 The vector combination of the Coriolis, pressure gradient, and dissipative terms is aligned down-fjord. Therefore, fluid parcels accelerate down-fjord. Figure 7 shows this as the total acceleration:

$$\frac{d\mathbf{u}}{dt} = \frac{\partial \mathbf{u}}{\partial t} + (\mathbf{u} \cdot \nabla)\,\mathbf{u}, \tag{9}$$

but the acceleration is dominated by the advective term. Note, for example, how the flow leaving the control volume in Figure 7 at $x = 3\,\mathrm{km}$ is faster than that entering at $x = 1\,\mathrm{km}$. At a fixed point the flow is
285 approximately constant.

Make analogy to wind at earth's surface, which is ageostrophic due to friction.

**4 Parameter space (working heading)**

A lot of stuff about melt is slightly off since I wasn't including plume melt. For $Q_{sg} = 100 \, \mathrm{m^3 \, s^{-1}}$, it adds about $2 \, \mathrm{m^3 \, s^{-1}}$ of melt

Thinking of doing another series of sensitivity tests with the minimum velocity value

The overturning circulation—with a thin, fast, surface outflow and a thick, slow outflow at depth— is dominated by the subglacial discharge plume. Meltwater coming from the icebergs has little effect, especially in the region down-fjord of the melange. At least, this is true for the subset of simulations in the previous section.

In the coming section, we test a wider parameter space to examine the influence of subglacial discharge, stratification, turbulent transfer at the ice–ocean interfaces, and the ambient water temperature. We focus on how these affect the outflow (evaluated just beyond the end of the melange) and the total ice melt rate.

Say something about default parameters

Say something about how much of simple plume scaling is already done in Slater et al. (2016) and probably elsewhere

To provide reference points for the magnitudes of $Q_{\mathrm{out}}$ as functions of the various parameters, we will include the values of the volume flux predicted using a simple half-conical plume model that has neither ice melt nor drag:

$$\frac{\mathrm{d}}{\mathrm{d}z}\left(r^2 w\right) = 2\alpha r w \tag{10}$$

$$\frac{\mathrm{d}}{\mathrm{d}z}\left(r^2 w^2\right) = g' r^2 \tag{11}$$

$$\frac{\mathrm{d}}{\mathrm{d}z}\left(r^2 w T_{\mathrm{p}}\right) = 2\alpha r w T_{\mathrm{a}}(z) \tag{12}$$

$$\frac{\mathrm{d}}{\mathrm{d}z}\left(r^2 w S_{\mathrm{p}}\right) = 2\alpha r w S_{\mathrm{a}}(z) \tag{13}$$

where $r$ is the half-conical plume's radius and $w$ its vertical velocity, $T$ and $S$ are temperature and salinity of the plume and ambient water (with subscripts p and a), $g'$ is the plume's reduced gravity ($g\Delta\rho/\rho_0$), and $\alpha = 0.1$ is the same entrainment coefficient used in the full plume parameterization.

We solve Equations 10–13 numerically with initial conditions of $w_0 = 1\,\mathrm{m\,s^{-1}}$ and $r_0 = \sqrt{2Q_\mathrm{sg}/\pi w_0}$. The plume's volume flux is

$$Q_\mathrm{p} = \pi r^2 w/2 \qquad (14)$$

where $r$ and $w$ are values when the plume either reaches the surface or neutral buoyancy. At this time, we assume the vertical volume flux turns 90° to become a horizontal volume flux.

Something about Sa, Ta are the same as the model initial conditions.

**4.1  Role of iceberg density ($\lambda$)**

To introduce Figure 8, we first revisit the simulations in which $\lambda$ varies from 0 to 0.3 (Figure 8a). Compared against other parameters to be discussed, $Q_\mathrm{out}$ is insensitive to $\lambda$, and the simple plume model explains 80% of $Q_\mathrm{out}$. This prediction is the $\lambda$-independent dashed line in Figure 8a. Loosely, this implies that the outflow's transit through and under the melange increases its magnitude by $\sim$20%. This increase occurs through the addition of meltwater and entrainment at the bottom of the outflow.

The total meltwater flux scales approximately linearly with $\lambda$. If we subtract the $4\,\mathrm{m^3\,s^{-1}}$ of melting at the glacier face, we find that $Q_\mathrm{melt} \propto \lambda^{0.90}$ and that $Q_\mathrm{melt} \propto A_\mathrm{ice}^{0.96}$ where $A_\mathrm{ice}$ is the total surface area of icebergs in contact with water. This scaling agrees with satellite(?) observations from Enderlin et al. (2018) who note that meltwater fluxes from icebergs can be reasonably approximately as a linear function of submerged ice area (Need to reread and check context of this statement).

**4.2  Role of subglacial discharge ($Q_\mathrm{sg}$)**

Subglacial discharge has a large, mostly predictable influence on outflow. For weaker discharge or stronger stratification, the plume reaches neutral buoyancy before the surface and $Q_\mathrm{out} \propto Q_\mathrm{sg}^n$ where $n \approx 0.75$ (e.g., Slater et al., 2016). For stronger discharge or weaker stratification, the plume reaches the surface and the exponent $n$ is approximately 0.37. Say something about n being 0.37 rather than theoretical 0.33. For our default, relatively weak stratification of 0.5 psu/600 m, we are in the $Q_\mathrm{sg}^{1/3}$ regime whenever $Q_\mathrm{sg} \gtrsim 10\,\mathrm{m^3\,s^{-1}}$.

Mention also Figure 3 of Xu et al. (2013).

[Figure]

**Figure 8.** Outflow and ice melt is sensitive to some model parameters and insensitive to others. Panel a melt numbers don't match Figure 6 because I'm using different snapshots (I think). This will be fixed in future.

Except for the small discharge runs ($Q_{sg} \leq 20\,\text{m}^3\,\text{s}^{-1}$), outflow is approximately 20% larger than predicted by the simple plume model—the same as for $\lambda$.

The total melt rate with no discharge is $13.4\,\text{m}^3\,\text{s}^{-1}$. This equates to an average melt rate of $11\,\text{cm}\,\text{d}^{-1}$ water equivalent spread over the $10^7\,\text{m}^2$ of ice ($\lambda = 0.1$). For all but 0.3% of this ice area, the current

340 speed in the adjacent cell is below the $4\,\text{cm}\,\text{s}^{-1}$ threshold (Equation 7). Hence, the total melt is controlled by the choice of this threshold that represents the effects of melt-driven convection.

I haven't calculated the above percentages exactly. I need to redo these and weight the average by the size of the ice-adjacent cells.

Remind reader somewhere that no-discharge simulation is shown in Figure 2

345 With increasing discharge, melt increases linearly. A fit across the five simulations from 0 to $200\,\mathrm{m}^3\,\mathrm{s}^{-1}$ gives

$$Q_{\mathrm{melt}} = 13.4 + 0.04 Q_{\mathrm{sg}} \qquad (15)$$

This is curiously similar to Figure 6 of Cenedese and Gatto (2016). Need to investigate further. By comparison, plume-driven melt at the glacier face is

350 $$Q_{\mathrm{melt}} = 0.4 Q_{\mathrm{sg}}^{0.38} \qquad (16)$$

which is an order of magnitude smaller than Equation 15.

This is empirical for the specific salinity stratification. Elaborate on what this means.

Maybe also compare to melt $\propto Q_R^{0.32}$ from Jackson et al. (2022) and $Q^{1/3}$(?) in Jenkins (2011).

**4.3    Role of salinity stratification ($\mathrm{d}S/\mathrm{d}z$)**

355 Outflow typically weakens with increasing salinity stratification (Figure 8c). Qualitatively, this makes sense given the plume's behavior: with increased stratification, the plume tends toward neutral buoyancy earlier, which slows it down and causes it to entrain less ambient fluid overall. Less entrainment during the plume's rise equates to a weaker outflow. This argument, however, does not explain the $\sim$40% reduction in $Q_{\mathrm{out}}$ as $\Delta S$ goes from 0.2 to 2.0. The simple plume model predicts only a 20% reduction.

360 In weak stratifications in which the plume reaches the surface, the outflow passes through the melange and moves slowly but surely down fjord as, for example, in Figure 4d. Conversely, outflow has a different fate if the plume flows out before reaching the surface. Consider a case in which the plume reaches neutral buoyancy at $\sim$100 m, which leads to the outflow interacting with far fewer icebergs. As per the momentum budget analysis in Section 3.3, this implies that the plume flow will tend toward geostrophic balance. In

365 this case, geostrophic balance manifests as an eddy in front of the glacier. In other words, the outflow at $x = 5\,\mathrm{km}$ is reduced because much of the flow recirculated back up fjord before reaching this point.

Some of this is probably wrong and based on an instability when the plume is in the exact center of the domain.

**4.4 Role of turbulent transfer coefficients ($\gamma_T, \gamma_S$)**

370 Increasing the ice–ocean turbulent transfer coefficients increases melt rates and outflow. Doubling the transfer coefficients nearly doubles the input of meltwater, and similarly for quadrupling. For these two scaling factor, the extra meltwater induces 7 and 17% more outflow, respectively.

We see hints of a positive feedback when we further increase the transfer coefficients. That is, increased turbulent transfer coefficients lead to more melt, which increases the speed of the surface outflow, which

375 increases the turbulent transfer through the dependence on speed (Equation 3). Specifically, using $8 \times \gamma_{T,S}$ leads to $8.4 \times$ the total melt.

This feedback is, of course, not large. Instead, to first order the melt rate should be considered linearly proportional to $\gamma_T$ and $\gamma_S$. The reason for this proportionality is that turbulent transfers to the ice–ocean interface are velocity dependence only if speeds exceed $0.04 \, \mathrm{m\,s^{-1}}$ (Equation 7). For $1\times, 2\times$, and $4 \times \gamma_{T,S}$,

380 only 24–26% of ice-adjacent cells exceed this threshold. For $8 \times \gamma_{T,S}$, this goes up, but only to 31%.

**4.5 Role of ambient temperature ($T_a$)**

This section will change a bit once I get

Ambient temperature has approximately linear influences on both outflow and melt rate. The effect on outflow is incremental: a $1°C$ increase leads to a $< 2\%$ increase to the otherwise plume-driven outflow.

385 The effect on melt rate, by comparison, is large. We find that total melt is proportional to $\Delta T^{1.1}$ where the thermal forcing $\Delta T = T - T_{\mathrm{fp}}$ with $T_{\mathrm{fp}} \approx -1.9°C$. If we consider only icebergs, the total melt scales as $\Delta T^{1.2}$.

These exponents of 1.1 and 1.2 are lower than might be expected. For a vertical ice face melting in a stratified environment without any discharge, Greisman (1979) predict that melt will scale as $\Delta T^{1.6}$, and

390 this prediction agrees with 1-m resolution simulations of an idealized Store glacier by Xu et al. (2013). The found that melt scaled as $\Delta T^{1.61}$ for simulations without subglacial discharge. Magorrian and Wells (2016) even find a depth-averaged melt rate proportional to $\Delta T^2$. (Need to check conditions under which this is valid. Might not apply here.)

**4.6 Realistic temperature and salinity**

395 Might be interesting to know if melt remains at depth due to near-surface pycnocline. And if we see more than a single overturning cell. Observations from some fjords show two inflow and two outflow layers (Inall et al., 2014; Mortensen et al., 2020), as do some models (Sciascia et al., 2013; Zhao, 2022).

These runs are also probably wrong on account of the instability with the plume in the middle. I therefore haven't written anything.

400 I should do the winter run without discharge

**5  Comparison with a sub-grid scale iceberg parameterization**

Resolving flow around icebergs is desirable, but bears a computational burden: simulating even a fraction of a fjord at 10-m resolution for a few days requires $\mathcal{O}(1000)$ CPU hours. This drops to $\mathcal{O}(0.1)$ CPU hours if we lower the resolution to, say, 400 m and parameterize (rather than resolve) icebergs. But does this
405 entail a loss of accuracy?

No, the loss of accuracy is minimal, as we will show.

**5.1 Setup of coarse simulations**

We rerun the same 22(?) simulations described in Section 4 at 400-m horizontal resolution with the Davison et al. (2020, 2022) iceberg scheme. Hereafter, we will refer to the original 10-m grid as the *fine grid*
410 and this new 400-m grid as the *coarse grid*.

The iceberg populations and dimensions are the same between the two grids. For the default value of $\lambda = 0.1$, a typical $400 \times 400$ m cell contains 3–5 icebergs. As for the fine grid, icebergs in the coarse simulations are stationary, and the same ice–ocean interface thermodynamics and turbulent transfer coefficients are used (see Section 2.2).
415 For the coarse grid, the subglacial plume fits within one cell, so no smoothing is used as it was for the fine grid (Section 2.3). Horizontal and vertical mixing schemes are the same between the different grids.

Do I want to say this this last point is probably a bad idea? But that it's a second-order effect, I think?

[Figure]

**Figure 9.** Comparing velocity profiles within and beyond the melange using two different approaches to simulating iceberg dynamics. Both models have the same population of icebergs. In the coarse-grid simulation, icebergs are sub-grid scale components. In the fine-grid simulation, icebergs are explicitly resolved and occupy volume. The two simulations use the default parameters ($\lambda = 0.1, Q_{sg} = 100\,\mathrm{m}^3\,\mathrm{s}^{-1}$).

**5.2 Coarse vs fine simulations**

Icebergs induce a blocking effect and thicken the outflow. Figure 6, for example, shows the outflow in the absence of icebergs as a 20-m thick surface-intensified jet, but with many icebergs it becomes thicker, slower, and peaks at $\sim 50\,\mathrm{m}$ depth. Beyond the melange, the blocking effect disappears and the jet becomes surface intensified.

The blocking effect does not occur when using the Davison et al. (2020) iceberg scheme. Instead, we see surface-intensified jets within the melange (Figure 9a) as well as beyond the melange (Figure 9b). Surface flows within the melange in the coarse simulations are up to two times too fast compared to the fine grid.

Davison et al. (2020) recognized that the icebergs act as barriers: one input to their scheme is a three-dimensional array that dictates the fraction of each grid cell occupied by ice. In principle, seawater should be squeezed through the remaining fractions of the cells. Our inspection and tests of the code, however, showed that the iceberg barrier mask is not implemented correctly and has no effect.

Check out testing of this hypothesis in idealised/iceberg_check_barrier_effect and idealised/melange_vs_iceberg_column

The lack of a blocking effect is not a major concern if we are interested in flow properties beyond the melange. Figure 9b shows that the average cross-channel velocity profiles just 500 m downstream of the melange agree reasonably well between the coarse and fine grids, as do the temperature and salinity (not shown). More generally, there is reasonable agreement between coarse and fine simulations across the 22 scenarios. The average discrepancy in outflow evaluated just down-fjord of the melange is less than 10% (Figure 10a). Similarly, the average discrepancy in total melt rate is less than 25% (Figure 10b).

These percentages definitely need to be recalculated correctly.

Given these good agreements, we recommend using the Davison et al. iceberg scheme for fjord-scale or larger models unless there is a specific goal to simulate vertical profiles of mass and freshwater fluxes. If not, then the iceberg scheme is sufficient to predict realistic fjord-wide distributions of sinks of heat and salt from melting icebergs—at least to within the known limitations of the turbulent transfer coefficients (see Section ?)

**6  Discussion**

Notes in no particular order

Jackson et al. (2020) shows that to get melt rates correct at LeConte, you could increase the drag coefficient by 175. Or, you could use much more modest increases of 4x(?) if you include the observed velocity of $0.2\,\mathrm{m\,s^{-1}}$. The implication then is that you want to understand the velocity in the vicinity of ice. Slater et al. (2018) showed how the circulation would double the melt of the glacier in Sarqardleq Fjord. They found ...

I'm not resolving the ambient melt plumes at the interface of the icebergs. These plumes will entrain water and thereby increase their outflow. The model is therefore likely underestimating how much of an

[Figure]

**Figure 10.** Coarse and fine grid simulations agree well in terms of (a) outflow and (b) meltwater flux. The coarse-grid simulations use the sub-grid scale iceberg parameterization from Davison et al. (2020) and are 40× coarser in the horizontal (400 m vs 10 m) than the fine-grid simulations described in the rest of the paper. Neither melt is calculated correctly. They'll both be higher than the values here.

effect icebergs have. At the same time, these ambient plumes don't rise up far before reaching neutral
buoyancy. So there is a series of weak intrusions than extend only a few hundred meters (Jackson et al.,
2020) rather than a few larger plumes. What do papers like Magorrian and Wells (2016) have to say about
this?

My results showed that the $0.04\,\mathrm{m\,s^{-1}}$ threshold used to approximate the effect of ambient plumes plays
a large role. This is more reason to get this ambient melt problem solved. Hager et al. (2022) set threshold
to 0.9 for summer and $0.1\,\mathrm{m\,s^{-1}}$ in winter. Josberger and Neshyba (1980) measured vertical velocities of
$0.07\,\mathrm{m\,s^{-1}}$ next to iceberg off Canada.

Also, I've yet to touch on the issue of how turbulent transfer coefficients should depend on model
resolution. Sciascia et al. (2013), for example, found that changing resolution from 10 to 50 m reduced
mean glacier melt rate by factor of 2, but that increasing to 5 m also lead to decrease in melt rate. Overall,
10 m gave highest rate out of 5, 10, 20, and 50 m. Xu et al. (2012) found ∼60% higher melt rates with

horizontal resolution of 2 m instead of 20 m. Slater et al. (2015) found 15% decrease in melt moving from 5 to 10 m. Maybe finite-element models are the answer, or at least a complementary approach (e.g. Kopera et al., 2022). According to Shin et al. (2021), you need 24 grid points per cube to resolve turbulent features at the edges of cubes.

470 As Malyarenko et al. (2020) put it, 'external velocity sources can significantly complicate the entrainment rate and the boundary layer structure next to the sloping interface'. The complex pathways through the melange can be thought of as something affecting the "external velocity sources"

I haven't dealt with forcing coming in from mouth of fjord, which would increase velocity variance.

Truffer and Motyka (2016) Section 5.3 notes that melange may well be blocking ascent of the plume.
475 Although my model isn't sufficient to prove that this blocking can occur, it does show that the plume prefers to flow out beneath the melange. Truffer and Motyka (2016) also point out that 'much speculation remains ... about the behavior and impact of submarine melting on the melange'. Also relevant to this point is **?**

Is it possible to use measured rates of melting across the melange from Enderlin to tune turbulent
480 transfer coefficients? It's hard to tune coefficients at the glacier face because it's hard to accurately measure it's melt rate. Maybe it's easier to use icebergs, which are floating and don't have an upstream source like a glacier does. In other words, the mass budget is easier.

Can I compare results to Hester et al. (2021) who looked at melting of one iceberg in a current vs not in a current.

485 What is depth distribution of meltwater among the simulations? As Moon et al. (2018) notes, large-scale models often simply add freshwater to surface. And does this have any implications for the work of Slater et al. (2022), who note that role of icebergs is missing

How does glacial meltwater fraction vary with distance down fjord? How does this compare to observations by Mortensen et al. (2020)?

490 Does plume thickness/melting agree at all with Everett et al. (2021)?

Obvious next step would be to introduce intermediary circulation a la Gladish et al. (2015). Feasible to look at that time scale and see what happens to the wave. Though not immediately obvious how to do it in 3D

**7 Conclusions**

*Code availability.* TEXT

*Data availability.* TEXT

**Appendix A: Spin up**

All simulations start from rest with horizontally homogeneous temperature and salinity. Plume-induced currents spread radially over the a few hours until reaching the fjord boundaries and then continue extending down-fjord thereafter. For the base case, currents reach the eastern end of the melange within 12 h (Figure A1). Currents speeds at this location averaged across the fjord keep increasing until $t \approx 24$ hr after which they are in an approximately steady state (solid line in Figure A2a). Temperature at the same location continues to drop at $t = 24$ h, which is to be expected given the distributed heat sink that is the melange. After 48 h, the rate of change of temperature is small (solid line in Figure A2b) because the heat loss from ice is largely balanced by a down-fjord flux of cooler water.

For the plume-only case, there are no icebergs acting as obstacles for the outflowing plume, so it is quick to reach an energetically steady state. There is cooling due to the input of subglacial discharge, but it is more gradual than in simulations with icebergs.

For the ice-only case, a steady state is not reached before the simulation ends. Currents keep increasing and temperatures keep decreasing as ice melts. Based on a coarse grid simulation (Section 5), we estimate that this particular scenario requires $\sim 100$ h for current speed to reach a steady value, but temperatures continue to drop at a nearly constant rate of $\sim 0.03°\mathrm{C\,d}^{-1}$ because the down-fjord flux of cool water is too weak to balance the heat lost to melting.

Need to explain Figure A2c

Say something about wanting to evaluate models all at same time, rather than each one's steady state. Maybe put this not in section 2.

[Figure]

**Figure A1.** Spin up of the surface velocity field for $\lambda = 0.1$, $Q_{sg} = 100\,\mathrm{m}^3\,\mathrm{s}^{-1}$. Should change the limits to show that, outside the melange, things aren't steady.

**A1**

*Author contributions.* TEXT

*Competing interests.* TEXT

520 *Disclaimer.* TEXT

*Acknowledgements.* TEXT

[Figure]

**Figure A2.** Simulations tend toward steady state within 48 hours, except for the less energetic ice-only case. (a) The quantity $\sqrt{u^2 + v^2 + w^2}$ averaged over the top 100 m of a cross section at the eastern end of the melange ($x = 4.6\,\mathrm{km}$). (b) As for panel a but for temperature. (c) The salinity difference at the same location. Caption could be much improved.

**References**

Adcroft, A., C. Hill, J.-M. Campin, J. Marshall and P. Heimbach (2004), Overview of the formulation and numerics of the MIT GCM, in *Proceedings of the ECMWF seminar series on Numerical Methods, Recent developments in numerical methods for atmosphere and ocean modelling*, pp. 139–149.

Bendtsen, J., J. Mortensen and S. Rysgaard (2015), Modelling subglacial discharge and its influence on ocean heat transport in Arctic fjords, *Ocean Dyn.*, 65, 1535–1546, https://doi.org/10.1007/s10236-015-0883-1.

Carroll, D., D. A. Sutherland, E. L. Shroyer, J. D. Nash, G. A. Catania and L. A. Stearns (2015), Modeling turbulent subglacial meltwater plumes: Implications for fjord-scale buoyancy-driven circulation, *J. Phys. Oceanogr.*, 45, 2169–2185, https://doi.org/10.1175/JPO-D-15-0033.1.

Carroll, D., D. A. Sutherland, E. L. Shroyer, J. D. Nash, G. A. Catania and L. A. Stearns (2017), Subglacial discharge-driven renewal of tidewater glacier fjords, *J. Geophys. Res. Oceans*, 122, 6611–6629, https://doi.org/10.1002/2017JC012962.

Cenedese, C. and V. M. Gatto (2016), Impact of a localized source of subglacial discharge on the heat flux and submarine melting of a tidewater glacier: A laboratory study, *J. Phys. Oceanogr.*, 46, 3155–3163, https://doi.org/10.1175/JPO-D-16-0123.1.

Cowton, T., D. Slater, A. Sole, D. Goldberg and P. Nienow (2015), Modeling the impact of glacial runoff on fjord circulation and submarine melt rate using a new subgrid-scale parameterization for glacial plumes, *J. Geophys. Res. Oceans*, 120, 796–812, https://doi.org/10.1002/2014JC010324.

Davison, B. J., T. Cowton, A. Sole, F. Cottier and P. Nienow (2022), Modelling the effect of submarine iceberg melting on glacier-adjacent water properties, *The Cryosphere*, 16, 1181–1196, https://doi.org/10.5194/tc-16-1181-2022.

Davison, B. J., T. R. Cowton, F. R. Cottier and A. J. Sole (2020), Iceberg melting substantially modifies oceanic heat flux towards a major Greenlandic tidewater glacier, *Nat. Commun.*, 11, 5983, https://doi.org/10.1038/s41467-020-19805-7.

De Andrés, E., J. Otero, F. Navarro, A. Promińska, J. Lapazaran and W. Walczowski (2018), A two-dimensional glacier–fjord coupled model applied to estimate submarine melt rates and front position changes of Hansbreen, Svalbard, *J. Glaciol.*, 64, 745–758, https://doi.org/10.1017/jog.2018.61.

De Andrés, E., D. A. Slater, F. Straneo, J. Otero, S. Das and F. Navarro (2020), Surface emergence of glacial plumes determined by fjord stratification, *The Cryosphere*, 14(6), 1951–1969, https://doi.org/10.5194/tc-14-1951-2020.

Enderlin, E. M., C. J. Carrigan, W. H. Kochtitzky, A. Cuadros, T. Moon and G. S. Hamilton (2018), Greenland iceberg melt variability from high-resolution satellite observations, *The Cryosphere*, 12, 565–575, https://doi.org/10.5194/tc-12-565-2018.

Enderlin, E. M., G. S. Hamilton, F. Straneo and D. A. Sutherland (2016), Iceberg meltwater fluxes dominate the freshwater budget in Greenland's iceberg-congested glacial fjords, *Geophys. Res. Lett.*, 43, 11287–11294, https://doi.org/10.1002/2016GL070718.

FitzMaurice, A., C. Cenedese and F. Straneo (2017), Nonlinear response of iceberg side melting to ocean currents, *Geophys. Res. Lett.*, 44, 5637–5644, https://doi.org/10.1002/2017GL073585.

Fraser, N. J., M. E. Inall, M. G. Magaldi, T. W. N. Haine and S. C. Jones (2018), Wintertime fjord-shelf interaction and ice sheet melting in Southeast Greenland, *J. Geophys. Res. Oceans*, 123, 9156–9177, https://doi.org/10.1029/2018JC014435.

Gade, H. G. (1979), Melting of ice in sea water: A primitive model with application to the antarctic ice shelf and icebergs, *J. Phys. Oceanogr.*, 9, 189–198, https://doi.org/10.1175/1520-0485(1979)009<0189:MOIISW>2.0.CO;2.

Gladish, C. V., D. M. Holland, A. Rosing-Asvid, J. W. Behrens and J. Boje (2015), Oceanic boundary conditions for Jakobshavn Glacier. Part I: Variability and renewal of Ilulissat Icefjord waters, 2001–14, *J. Phys. Oceanogr.*, 45, 3–32, https://doi.org/10.1175/JPO-D-14-0044.1.

Greisman, P. (1979), On upwelling driven by the melt of ice shelves and tidewater glaciers, *Deep Sea Res. A*, 26, 1051–1065, https://doi.org/10.1016/0198-0149(79)90047-5.

Hager, A. O., D. A. Sutherland, J. M. Amundson, R. H. Jackson, C. Kienholz, R. J. Motyka and J. D. Nash (2022), Subglacial discharge reflux and buoyancy forcing drive seasonality in a silled glacial fjord, *J. Geophys. Res. Oceans*, 127, https://doi.org/10.1029/2021JC018355.

Hermann, A. J., P. B. Rhines and E. R. Johnson (1989), Nonlinear Rossby adjustment in a channel: beyond Kelvin waves, *J. Fluid Mech.*, 205, 469–502, https://doi.org/10.1017/S0022112089002119.

Hester, E. W., C. D. McConnochie, C. Cenedese, L.-A. Couston and G. Vasil (2021), Aspect ratio affects iceberg melting, *Phys. Rev. Fluids*, 6, 023802, https://doi.org/10.1103/PhysRevFluids.6.023802.

Hughes, K. G. (2022), Pathways, form drag, and turbulence in simulations of an ocean flowing through an ice mélange, *J. Geophys. Res. Oceans*, 127, e2021JC018228, https://doi.org/10.1029/2021JC018228.

Inall, M. E., T. Murray, F. R. Cottier, K. Scharrer, T. J. Boyd, K. J. Heywood and S. L. Bevan (2014), Oceanic heat delivery via Kangerdlugssuaq Fjord to the south-east Greenland ice sheet, *J. Geophys. Res. Oceans*, 119, 631–645, https://doi.org/10.1002/2013JC009295.

Jackson, R. H., R. J. Motyka, J. M. Amundson, N. Abib, D. A. Sutherland, J. D. Nash and C. Kienholz (2022), The relationship between submarine melt and subglacial discharge from observations at a tidewater glacier, *J. Geophys. Res. Oceans*, 127, e2021JC018204, https://doi.org/10.1029/2021JC018204.

Jackson, R. H., J. D. Nash, C. Kienholz, D. A. Sutherland, J. M. Amundson, R. J. Motyka, D. Winters, E. Skyllingstad and E. C. Pettit (2020), Meltwater intrusions reveal mechanisms for rapid submarine melt at a tidewater glacier, *Geophys. Res. Lett.*, 47, e2019GL085335, https://doi.org/10.1029/2019GL085335.

Jackson, R. H., E. L. Shroyer, J. D. Nash, D. A. Sutherland, D. Carroll, M. J. Fried, G. A. Catania, T. C. Bartholomaus and L. A. Stearns (2017), Near-glacier surveying of a subglacial discharge plume: Implications for plume parameterizations, *Geophys. Res. Lett.*, 44, 6886–6894, https://doi.org/10.1002/2017GL073602.

Jackson, R. H. and F. Straneo (2016), Heat, salt, and freshwater budgets for a glacial fjord in Greenland, *J. Phys. Oceanogr.*, 46, 2735–2768, https://doi.org/10.1175/JPO-D-15-0134.1.

Jenkins, A. (2011), Convection-driven melting near the grounding lines of ice shelves and tidewater glaciers, *J. Phys. Oceanogr.*, 41, 2279–2294, https://doi.org/10.1175/JPO-D-11-03.1.

Josberger, E. G. and S. Neshyba (1980), Iceberg melt-driven convection inferred from field measurements of temperature, *Ann. Glaciol.*, 1, 113–117, https://doi.org/10.3189/S0260305500017080.

Klymak, J. M. and S. M. Legg (2010), A simple mixing scheme for models that resolve breaking internal waves, *Ocean Modell.*, 33, 224–234, https://doi.org/10.1016/j.ocemod.2010.02.005.

Kopera, M., Y. Gahounzo, E. M. Enderlin and F. X. Giraldo (2022), Non-hydrostatic unified model of the ocean with application to ice/ocean interaction modeling, https://doi.org/10.13140/RG.2.2.35929.80488.

Losch, M. (2008), Modeling ice shelf cavities in a $z$ coordinate ocean general circulation model, *J. Geophys. Res.*, 113, C08043, https://doi.org/10.1029/2007JC004368.

Magorrian, S. J. and A. J. Wells (2016), Turbulent plumes from a glacier terminus melting in a stratified ocean, *J. Geophys. Res. Oceans*, 121, 4670–4696, https://doi.org/10.1002/2015JC011160.

Malyarenko, A., A. J. Wells, P. J. Langhorne, N. J. Robinson, M. J. Williams and K. W. Nicholls (2020), A synthesis of thermodynamic ablation at ice–ocean interfaces from theory, observations and models, *Ocean Modell.*, 154, 101692, https://doi.org/10.1016/j.ocemod.2020.101692.

Marshall, J., A. Adcroft, C. Hill, L. Perelman and C. Heisey (1997), A finite-volume, incompressible Navier Stokes model for studies of the ocean on parallel computers, *J. Geophys. Res.*, 102, 5753–5766, https://doi.org/10.1029/96JC02775.

610 Melton, S. M., R. B. Alley, S. Anandakrishnan, B. R. Parizek, M. G. Shahin, L. A. Stearns, A. L. LeWinter and D. C. Finnegan (2022), Meltwater drainage and iceberg calving observed in high-spatiotemporal resolution at Helheim Glacier, Greenland, *J. of Glaciology*, 68(270), 812–828, https://doi.org/10.1017/jog.2021.141.

Moon, T., D. A. Sutherland, D. Carroll, D. Felikson, L. Kehrl and F. Straneo (2018), Subsurface iceberg melt key to Greenland fjord freshwater budget, *Nat. Geosci.*, 11, 49–54, https://doi.org/10.1038/s41561-017-0018-z.

615 Mortensen, J., S. Rysgaard, J. Bendtsen, K. Lennert, T. Kanzow, H. Lund and L. Meire (2020), Subglacial discharge and its down-fjord transformation in West Greenland fjords with an ice mélange, *J. Geophys. Res. Oceans*, 125, e2020JC016301, https://doi.org/10.1029/2020JC016301.

Moyer, A. N., D. A. Sutherland, P. W. Nienow and A. J. Sole (2019), Seasonal variations in iceberg freshwater flux in Sermilik Fjord, Southeast Greenland from Sentinel-2 imagery, *Geophys. Res. Lett.*, 46, 8903–8912, 620 https://doi.org/10.1029/2019GL082309.

Rezvanbehbahani, S., L. A. Stearns, R. Keramati, S. Shankar and C. J. van der Veen (2020), Significant contribution of small icebergs to the freshwater budget in Greenland fjords, *Commun. Earth Environ.*, 1, 31, https://doi.org/10.1038/s43247-020-00032-3.

Schulz, K., A. T. Nguyen and H. R. Pillar (2022), An improved and observationally-constrained melt rate pa- 625 rameterization for vertical ice fronts of marine terminating glaciers, *Geophys. Res. Lett.*, 49, e2022GL100654, https://doi.org/10.1029/2022GL100654.

Sciascia, R., F. Straneo, C. Cenedese and P. Heimbach (2013), Seasonal variability of submarine melt rate and circulation in an East Greenland fjord, *J. Geophys. Res. Oceans*, 118, 2492–2506, https://doi.org/10.1002/jgrc.20142.

Shin, H. H., D. Muñoz-Esparza, J. A. Sauer and M. Steiner (2021), Large-eddy simulations of stability- 630 varying atmospheric boundary layer flow over isolated buildings, *J. Atmos. Sci.*, 78, 1487–1501, https://doi.org/10.1175/JAS-D-20-0160.1.

Slater, D. A., D. Carroll, H. Oliver, M. J. Hopwood, F. Straneo, M. Wood, J. K. Willis and M. Morlighem (2022), Characteristic depths, fluxes, and timescales for Greenland's tidewater glacier fjords from subglacial discharge-driven upwelling during summer, *Geophys. Res. Lett.*, 49, e2021GL097081, 635 https://doi.org/10.1029/2021GL097081.

Slater, D. A., D. N. Goldberg, P. W. Nienow and T. R. Cowton (2016), Scalings for submarine melting at tidewater glaciers from buoyant plume theory, *J. Phys. Oceanogr.*, 46, 1839–1855, https://doi.org/10.1175/JPO-D-15-0132.1.

Slater, D. A., P. W. Nienow, T. R. Cowton, D. N. Goldberg and A. J. Sole (2015), Effect of near-terminus subglacial hydrology on tidewater glacier submarine melt rates, *Geophys. Res. Lett.*, 42(8), 2861–2868, https://doi.org/10.1002/2014GL062494.

Slater, D. A., F. Straneo, S. B. Das, C. G. Richards, T. J. W. Wagner and P. W. Nienow (2018), Localized plumes drive front-wide ocean melting of a Greenlandic tidewater glacier, *Geophys. Res. Lett.*, 45, 12350–12358, https://doi.org/10.1029/2018GL080763.

Straneo, F., R. G. Curry, D. A. Sutherland, G. S. Hamilton, C. Cenedese, K. Våge and L. A. Stearns (2011), Impact of fjord dynamics and glacial runoff on the circulation near Helheim Glacier, *Nat. Geosci.*, 4, 322–327, https://doi.org/10.1038/ngeo1109.

Stuart, G. A., M. A. Sundermeyer and D. Hebert (2011), On the geostrophic adjustment of an isolated lens: Dependence on Burger number and initial geometry, *J. Phys. Oceanogr.*, 41, 725–741, https://doi.org/10.1175/2010JPO4476.1.

Sulak, D. J., D. A. Sutherland, E. M. Enderlin, L. A. Stearns and G. S. Hamilton (2017), Iceberg properties and distributions in three Greenlandic fjords using satellite imagery, *Ann. Glaciol.*, 58, 92–106, https://doi.org/10.1017/aog.2017.5.

Truffer, M. and R. J. Motyka (2016), Where glaciers meet water: Subaqueous melt and its relevance to glaciers in various settings, *Rev. Geophys.*, 54, 220–239, https://doi.org/10.1002/2015RG000494.

Wood, M., E. Rignot, I. Fenty et al. (2021), Ocean forcing drives glacier retreat in Greenland, *Sci. Adv.*, 7, eaba7282, https://doi.org/10.1126/sciadv.aba7282.

Xu, Y., E. Rignot, I. Fenty, D. Menemenlis and M. M. Flexas (2013), Subaqueous melting of Store Glacier, west Greenland from three-dimensional, high-resolution numerical modeling and ocean observations, *Geophys. Res. Lett.*, 40, 4648–4653, https://doi.org/10.1002/grl.50825.

Xu, Y., E. Rignot, D. Menemenlis and M. Koppes (2012), Numerical experiments on subaqueous melting of Greenland tidewater glaciers in response to ocean warming and enhanced subglacial discharge, *Ann. Glaciol.*, 53, 229–234, https://doi.org/10.3189/2012AoG60A139.

Yankovsky, A. E. and I. Yashayaev (2014), Surface buoyant plumes from melting icebergs in the Labrador Sea, *Deep Sea Res.*, 91, 1–9, https://doi.org/10.1016/j.dsr.2014.05.014.

Zhao, K. (2022), Standing eddies in glacial fjords and their role in fjord circulation and melt, *Earth and Space Science Open Archive*, https://doi.org/10.1002/essoar.10511100.1.

Zhao, K. X., A. L. Stewart and J. C. McWilliams (2021), Geometric constraints on glacial fjord–shelf exchange, *J. Phys. Oceanogr.*, 51, 1223–1246, https://doi.org/10.1175/JPO-D-20-0091.1.

670  Zhao, K. X., A. L. Stewart and J. C. McWilliams (2022), Linking overturning, recirculation, and melt in glacial fjords, *Geophys. Res. Lett.*, 49, e2021GL095706, https://doi.org/10.1029/2021GL095706.

---

## Author Comment (AC2)

**Response to reviewers**

Hughes (2023) Fjord circulation induced by melting icebergs, doi:10.5194/egusphere-2023-2106

Thank you to both reviewers for the careful reading of the paper and encouraging feedback.

My responses to Reviewer 2 begin on page 7 of this document.

**Reviewer 1**

Review of "Fjord circulation induced by melting icebergs", by Kenneth G. Hughes, (egusphere-2023-2106)

The manuscript describes the development and application of an analytical model of fjord circulation forced by meltwater input at the surface (the meltwater stems from icebergs in the fjord). The build process of the model is described in an easy to follow fashion by considering and assembling different configurations with increasing complexity. The solutions agree reasonably well (considering the approximations implicit to the analytical solution) with high-resolution numerical simulations. A parameter study shows expected and not so intuitive results that can explain some observed behaviour in glacial fjords. The manuscript is well written and easy to follow.

Thanks for your review

As my main critique, the manuscript introduction does not describe the purpose nor the context of the study, so that the manuscript appears to be "just" a model description ("The core contribution of this paper is an analytical model explaining the first-order dynamics of a fjord's response to hundreds of melting icebergs.") as a very well designed and carried out exercise of geophysical fluid dynamics. In the same way the conclusion is a useful and alternative summary of the model steps, but no geophysical/scientific conclusion are drawn. Adding context and purpose would make this manuscript an even better paper. That's why I recommend minor revisions.

The impetus for this paper was a suite of numerical simulations that I began a couple of years ago that were much like those described in the paper, but that also included subglacial discharge. In trying to make sense of those simulations, I realized that there is a big discrepancy between our thorough knowledge on the role of subglacial discharge and our dearth of knowledge on the role of iceberg melt when it comes to explaining fjord dynamics. To me, there was a clear need in the literature for a paper that explains the mechanism by which iceberg melt is translated into currents. I have added a paragraph to the Introduction that expresses this discrepancy and therefore provides context for my study.

I also agree that the Conclusion was missing a statement on the consequences/significance of the work and that it ended in a weak way. I have taken what was the opening paragraph of Section 5.2 and moved it to the end of the Conclusion where it will be more visible. This paragraph about the potential for the analytical model to be extended and then applied to fjords across Greenland is, in my opinion, the most relevant high-level implication of the paper.

Not as important:

The model description uses some formulae, but it is not always clear (to me) where they come from, or how they were derived. I appreciate the briefness and clarity of the presentation, but the presentation requires that the reader "believes" the text. Maybe more information in an appendix would help?

Reviewer 2 had a similar concern. As I also state in response to their comments, there are several expressions in this paper that are simple yet that result from elaborate derivations that I want to avoid repeating since the details are tangential. I now point the reader to places where they can find more details if they please.

There are some technicalities, questions, and notes from reading the manuscript are listed below:

Abstract: no motivation, no concluding remarks about implications

At a guess, I'm inferring that the reviewer is suggesting a motivating statement or two that touches on, say,

- the role fjords have in being gateways for heat to the Greenland ice sheet;
- the role icebergs have in modifying glacier-adjacent water properties; or
- processes that are missing in climate models.

Such statements are appropriate if the paper has an expected audience than spans a range of scientific fields. However, I foresee my paper being read primarily by physical oceanographers and glaciologists: people who will already be familiar with its broad purpose. Therefore, I prefer to get straight to the point and keep the abstract concise.

*At the moment* there are few obvious geophysical implications of my work. As I and others build on the work and start invoking the analytical model in more applied studies, then there will be more tangible implications. But at this point, I am hesitant to conjecture in the abstract about what these implications will be.

l2: "Down-fjord" Is this a proper term? I would use downstream.

*Down-fjord* is the best term here and a google scholar search shows many uses of the term in other papers. *Downstream* could be an ambiguous term since the current direction changes with depth.

l24-25 "over the top 200 m is a few cm/s over the top 200 m"
repetition, also I would not associate "outflow" with velocity units. Maybe "outflow velocity"?

Wording fixed as suggested.

l27: this paragraph is very different in style from all other paragraphs, consider rewriting

This style of this paragraph is intentional. I want to emphasize that this is a process study that will address a parameter space. The four short questions in a row is a concise way to evoke this idea.

This idea of listing questions provides a nice transition to the following paragraph where I state what I plan to do to answer those questions.

page 4: The icebergs do not seem to move. Is that justified? Later this is described, but it may be useful to do it here already.

I have reworded part of the section in question as follows:

*Each iceberg is assumed to be a stationary cuboid. Although there are limitations with this approach, it is their cumulative meltwater flux, rather than their movement, that matters most here. Further detail and justification of our approach is given by Hughes (2022).*

l64: (practical) salinity has no units, as it is a ratio of g salt / kg sea water, see UNESCO reports etc. "psu" should not be used (I know that is done commonly, but it's still not correct). Absolute salinity (according to TEOS10) has units of g/kg.

I have removed the units. Personally, I have no strong opinion either way on whether they should be there or not. I am not using g/kg units because the equation of state that I'm using in the MITgcm predates TEOS10.

l73: At this resolution, non-hydrostatic dynamics may start to become important. Why use hydrostatic dynamics?

Non-hydrostatic dynamics are most needed when one is interested in accurately simulating the dynamics of overturning structures. This is not the case for my setup, which features a very gradual forcing.

The one place where non-hydrostatic dynamics will matter is beside the ice—where buoyant meltwater rises vertically—but I'm not going to come close to resolving that at 10-m resolution. This unresolved physics beside the ice–ocean interface is a whole different scientific question that is way beyond the scope of my paper.

I did run a few simulations on a smaller domain with non-hydrostatic dynamics included. The difference in the solutions was negligible for the purposes of this paper. Any improvement in accuracy is not worth the 2–3× slowdown in wall time needed for the simulations.

l77: "(designated as scheme 33 in the MITgcm)", I think the reference for this DST (direct space time) scheme is

Hundsdorfer, W. and Trompert, R. A.: Method of lines and direct discretization: a comparison for linear advection, Appl. Numer. Math., 13, 469–490, doi:10.1016/0168-9274(94)90009-4, 1994.

Hundsdorfer, W., Koren, B., van Loon, M., and Verwer, J.: A positive finite-difference advection scheme, J. Comp. Phys., 117, 35–46, doi: 10.1006/jcph.1995.1042, 1995.

I have added the Hundsdorfer et al. (1995) reference

l91, are the initial conditions (especially T=2deg) reasonable? The high melt rate will probably reduce with lower ocean temperatures (and it does according to the parameter study later on).

The choice of temperature is reasonable. As noted in Section 2.1, the temperature being 4°C above the freezing point is "an average thermal forcing for Greenland fjords (Wood et al. 2021)".

The temperature that I use for the simulations is somewhat arbitrary because melt rate also depends on the (poorly constrained) turbulent transfer coefficients at the ice–ocean interface. Hence, inaccuracy in melt rate is dominated by the uncertainty in $\gamma_T$ and $\gamma_S$ (and their velocity dependence), not by temperature.

Ultimately, I focused on choosing a combination of parameters ($\theta_a$, $\gamma_T$, and $\gamma_S$) that give iceberg melt rates that are comparable to remotely sensed observations as noted in Section 2.2.

l97: only small -> only over a small? Fixed

l99: the exponential function implies Kelvin waves, why should they be the only ones? What about Poincare waves (mentioned in the text) and Rossby waves?

The effects of Poincare waves can safely be ignored because they are much smaller than Kelvin waves. This can be seen in Figure 4a: the Poincare waves extending across the channel are only just visible in the second panel and are non-existent in the panels below. It can also be seen in Figure 8c, where I needed to annotate the Poincare waves because otherwise they would be easily missed.

Rossby waves will not occur in this system. Conventional Rossby waves can't occur here because Coriolis frequency is constant, and topography Rossby waves can't occur here because there isn't sloping topography.

I reworded the opening paragraph of Section 3.1 to make clear that Kelvin waves are the dominant process. Poincare waves are mentioned for completeness.

Figure 2, it took me a while to understand this complicated figure. I can appreciate it's value, but maybe extend the description in the caption to say that the vertical slices are at the dashed lines in the top view (or put them, where they belong?). The "pseudo"-realistic perspective implies something different and confused me (e.g. I thought that (c) and (d) form the sides of the displayed domain of (b) and it took me a little to figure out that (b) is only part of (a), etc.)

I agree that this depiction was confusing. I have revised the figure so that it is now intuitive how the three slices relate to each other in space. Below is a figure summarizing the change I made.

[Figure]

[Figure]

l115: typo in "$x$— —$z$ domain"? Fixed

l142: "overkill" maybe too colloquial?

I have left this as is. Personally, I don't see issues with using colloquial language in scientific papers provided that it is clear what is meant.

l145: "For an infinitely narrow channel, the tanh term goes to zero", not clear why. Eq5 and 7 do not have any $x$ in them that can got infinity.

Channel width is the $y$ direction. It is $W$ that goes to zero. I have clarified this in the text: "For an infinitely narrow channel ($W \to 0$), ..."

Section 3.2 is physically plausible but no formulae support it.

I fail to see how adding math/formulae to this section would help. Geometrically the argument is simple.

My guess is that the reviewer is questioning one of two processes. Either (1) that the Kelvin wave maintains its shape while turning the corners or (2) that destructive interference leads to complete cancellation of the Kelvin waves. I was also skeptical at one point that this would be the case. But the comparison of the analytical and numerical models in Section 4 shows that these two processes must be happening.

Further, although it is not noted in the paper, Figure 4 was created using MITgcm simulations with sea surface height varying in time just as described in Section 3.1–3.3. The fact that the results in Section 3.2 arise in a primitive equation model is good evidence that the process described can be trusted.

page 12

The figure needs more explanation, e.g. there's density in a+e, but pressure (anomaly) in b-d, but the caption doesn't describe this, nor the labels in the figure, …

I have added panel descriptions to the caption.

l200: Figure 4 -> Figure 5? Fixed

l209: "where z* is a dummy variable used to avoid ambiguity with the integral's lower limit". I guess that's common practise and need not be described. I have seen usually z' used for this.

I agree that it is common practice, especially if I were using the *z'* notation. However, I'm already using primes a lot in this paper to denote anomalies, so I want to avoid prime notation here. I decided that z* was the next best option, and then added the clarifying phrase in question to ensure there was no confusion as to what I was doing.

Eq18: integral over z? dz is missing. Not clear. Good catch. Fixed

Fig6 caption: relatively low-mode components. -> relatively larger low-mode components?

I now simply say "lower-mode" instead of the awkward "relatively low-mode"

page 17

l254: "The integer n_E is the maximum n" and the higher modes all destructively interfere?

No. The modes with $n > n_E$ are those that are too slow to have reached a given location $x$. The destructive interference is for modes $n \leq n_W$. Per a suggestion from reviewer 2, I have made this clearer in Equation 25.

page 23

l366: "by looking at" -> by evaluating (colloquial?) Fixed as suggested

ll370 maybe it would be a good idea to add the stationary role of the iceberg and neglected drag already in the model setup description? (See earlier comment on page 4)

Added as suggested

page 24

l380 Please discuss why the overestimation is reduced for the stronger stratification and weaker melt cases.

I have added the following sentence:
In fact, the agreement is better in these two scenarios compared to the default settings because the relative role of linear wave dynamics (compared to nonlinear advection) is larger when the outflow is weaker or the stratification is stronger.

page 29

l643: helps brings -> helps bring? Fixed

**Reviewer 2**

The author compares an analytic linearized fjord circulation model in the presence of an iceberg melange with numerical model simulation. I like this work and urge the author to keep refining their analytical model. I am not sure whether it will find applications in climate models; however, at the very least, it is a valuable tool to understand the outputs of numerical simulations.

Thank you for the review and encouragement to continue with this analytical approach. I agree that there's not an obvious way to add it to a climate model, and instead that its value is to help interpret numerical simulations.

I have several comments about the mathematical exposition of the author's work. I do not doubt the correctness, but the model derivation (being analytical) lacks certain rigor (or perhaps I lack understanding of the author's process, either of which merits further explanation).

Thanks for the suggestions/clarifications below. To develop the analytical model, I relied a lot on existing results. My contribution was effectively to repackage and combine various results, and apply some simple geometrical arguments so as to solve a new problem. In some cases, like Equations 11 and 19 in your comments 2 and 3, the simple expressions result from elaborate derivations that I want to avoid repeating since the details are tangential. I now point the reader to places where they can find more details if they please.

1. In sub-sections 3.x I am unsure whether the author follows Hermann 1989 or derives original equations. Please specify which part summarizes previous work and which is new. Most of this section reads as a recap of other work, but perhaps the exposition lacks detail. The following points provide specific questions stemming from this concern:

Hermann et al. (1989) technically only dealt with the open channel, abrupt release case in Section 3.1. I developed the generalizations in Section 3.2 and 3.3. However, these generalizations are simple geometrical arguments.

I now clarify in Section 3.2 what Hermann et al. (1989) did and didn't do.

2. How does the author derive relation (11)? Is this taken from another paper? If so, which one? Is N (which the author refers to as stratification) the same as buoyancy frequency in equation (3)?

I now point the reader to either the Kelly et al. (2010) or Gill (1982) for details on where this particular expression comes from. And I have changed the word "stratification" to "buoyancy frequency" to avoid confusion.

3. I do not understand how the author came up with equation (19). It is presented as a statement, not an assumption, and without much justification. Yet, the derivation of Fourier coefficients for velocity depends on it, so please explain more.

Again, Kelly et al. (2010) proves to be a good reference for this expression. Specifically, compare their Equations A12 and A14 with $\omega = 0$. I now note this in the text.

4. The author comments that they derive equations (23) and (24) from (21) and (22) by x-integration - please show how this happens, as I do not see the relation clearly. If the factor 2L comes from the integration, then are the terms in (21) and (22) space-independent? Also, integrated energy will not have the same unit as the original, so perhaps a different notation is required.

In hindsight, I agree my description of this step was too terse. It was also presented in an awkward order with both equations presented first and then described afterward in a single paragraph. I have revised the wording and dealt with each expression individually. It is now easier to follow the derivation.

And the reviewer is correct about units. I now write Equations 23 and 24 as $\int E_p$ and $\int E_k$ instead of just $E_p$ and $E_k$.

5. In Equation 25 the author subtracts two sums from each other. As stated, the result of the operation is in my mind is the sum of terms between $n_E$ and $n_W$ as the summation terms $u_n(z)$ are identical in both sums. Perhaps differentiate the notation here to indicate that terms in the $n_E$ sum are different than in the $n_W$ sum.

The reviewer is correct in interpreting the difference of the summations. I have added a second, equivalent expression showing that it is a sum of modes from $n_W+1$ to $n_E$.

6. Please include the numerical values of velocity in the figures comparing analytical and numerical results (i.e. Fig. 7, 8, etc). This is important for reproducibility.

I appreciate the suggestion to make the work reproducible, but I don't think that adding the numerical values to Figures 7 and 8 is the best way to do that because the actual current speeds are not the focus of those figures. Instead, I have uploaded scripts to recreate the results from Figures 7 and 8 to the Github repo and the Zenodo archive. These scripts produce velocity fields that *do* include numerical values.

7. Are the equations (32-35) justifiable in the light of the assumption of small perturbations made in Section 3.3?

In Section 3.3, I assume a *constant, gradual input of buoyant water*. In my opinion, this is an accurate description of the effects of ice melting in the ocean, which is what is being calculated in Equations 32–34. Of course, this doesn't necessarily imply that my assumptions are justifiable *a priori*. But the good agreement between the analytical and numerical models in Section 4 is evidence that the assumptions are valid.

8. It would be interesting to see the results for x<15 to get an idea of how far off the results are from the simulation. Perhaps including results at several locations (including further downstream than 20km) would increase the reader's sense of confidence in where the model can be applied, and where it fails.

I have added a second panel to Figure 11 that shows the total outflow for $0 < x < 100$ km at $t = 7$ days. This panel shows that the analytical and numerical models agree reasonably well provided $x > 20$ km, but diverge for smaller distances.